# A genetic selection reveals functional metastable structures embedded in a toxin-encoding mRNA

**Sara Masachis[†], Nicolas J Tourasse, Claire Lays, Marion Faucher, Sandrine Chabas, Isabelle Iost, Fabien Darfeuille***

University of Bordeaux, INSERM U1212, CNRS UMR 5320, ARNA Laboratory, Bordeaux, France

**Abstract** Post-transcriptional regulation plays important roles to fine-tune gene expression in bacteria. In particular, regulation of type I toxin-antitoxin (TA) systems is achieved through sophisticated mechanisms involving toxin mRNA folding. Here, we set up a genetic approach to decipher the molecular underpinnings behind the regulation of a type I TA in *Helicobacter pylori*. We used the lethality induced by chromosomal inactivation of the antitoxin to select mutations that suppress toxicity. We found that single point mutations are sufficient to allow cell survival. Mutations located either in the 5' untranslated region or within the open reading frame of the toxin hamper its translation by stabilizing stem-loop structures that sequester the Shine-Dalgarno sequence. We propose that these short hairpins correspond to metastable structures that are transiently formed during transcription to avoid premature toxin expression. This work uncovers the co-transcriptional inhibition of translation as an additional layer of TA regulation in bacteria.
DOI: https://doi.org/10.7554/eLife.47549.001

**\*For correspondence:**
fabien.darfeuille@inserm.fr

**Present address:** [†]Department of Microbiology, Faculty of Biology I, Ludwig-Maximilians-University of Munich, Martinsried, Germany

**Competing interests:** The authors declare that no competing interests exist.

## Introduction

In any living cell, unwanted gene expression can have a detrimental effect on cell growth, and eventually lead to cell death. In bacteria, a fine tuning of gene expression can be achieved at the translational level through the control of the accessibility of the ribosome binding site (RBS), which encompasses the Shine-Dalgarno (SD) sequence and the start codon. Messenger RNA structures occluding the RBS have been reported to control the expression of many important genes for which a timely control is crucial. Regulation of translation initiation via SD-sequestration is an old theme that initially started with the study of bacteriophage genes (*de Smit and van Duin, 1990*) and ribosomal proteins (for a review see *Duval et al., 2015*). More recently, its impact on other bacterial genes such as sigma factors (*Mearls et al., 2018*) and translational riboswitches (*Rinaldi et al., 2016*) has been reported.

Hence, in many cases, preventing gene expression via SD-sequence sequestration is crucial. This is particularly true for type I toxin-antitoxin (TA) systems. In contrast to the largest type II TA family, antitoxins belonging to type I TA systems do not directly interact with the toxin protein but rather prevent its expression (*Harms et al., 2018*). This regulation occurs through the direct base-pairing of the RNA antitoxin with the toxin mRNA and leads to toxin translation inhibition and/or mRNA degradation (*Brantl and Jahn, 2015*; *Durand et al., 2012*; *Wen and Fozo, 2014*). However, the action of the RNA antitoxin is often not sufficient to avoid toxin expression (*Masachis and Darfeuille, 2018*). Indeed, due to the coupling between transcription and translation occurring in bacteria, translation of the nascent toxin mRNA can potentially occur before antitoxin action. Thus, a tight control of toxin synthesis is usually achieved via the direct sequestration of its SD sequence within stable stem-loop structures (*Masachis and Darfeuille, 2018*). The existence of a non-translatable

toxin primary transcript is a major hallmark of type I TA loci. In this transcript, the RBS occlusion occurs through the base-pairing of the SD sequence with a partially or totally complementary sequence termed anti-SD (aSD). Such aSD sequences are often located a few nucleotides (up to 11) upstream or downstream of the SD sequence and trap it within a hairpin structure. However, in some cases, RBS occlusion involves an aSD sequence encoded far downstream and is achieved via a long-distance interaction (LDI) between the 5' and 3' ends of the toxin mRNA (*Gultyaev et al., 1997*; *Han et al., 2010*).

This strategy of toxin expression regulation via the formation of a LDI has been recently described for a type I TA family of the *Epsilon* proteobacteria. This family, named *aapA*/IsoA, is present in several copies on the chromosome of the major human gastric pathogen *Helicobacter pylori*. We characterized the *aapA1*/IsoA1 TA system at the locus I and showed that the *aapA1* gene codes for a small toxic protein whose expression is repressed by a *cis*-overlapping antisense RNA, IsoA1 (*Arnion et al., 2017*). We have shown that transcription of this toxic gene generates a highly stable primary transcript whose translation is post-transcriptionally impeded by a 5'−3' LDI (*Figure 1—figure supplement 1*). Consequently, a 3'-end ribonucleolytic event, that we termed 'mRNA activation step', is necessary to remove the LDI, thus enabling toxin translation (*Arnion et al., 2017*). We also showed that this peculiar mRNA folding is strongly conserved at various loci in other related bacterial species, clearly suggesting that this particular mode of control would also be conserved. However, several aspects of this regulation remain enigmatic. In particular, how can such a long-distance interaction prevent translation while the mRNA is being made? Indeed, in principle, the ribosome could bind to the nascent transcript, initiating toxin translation before the 3' end is synthesized. Previous work has shown that in some cases, metastable structures are formed during mRNA transcription to prevent ribosome binding and avoid translation (*Lai et al., 2013*; *Zhu and Meyer, 2015*).

In the present work, we aimed at deciphering further the mechanism of toxin regulation, studying another member of the *aapA*/IsoA family, *aapA3*/IsoA3. We first show that, in the absence of antitoxin expression, chromosomal toxin expression is lethal. Taking advantage of this lethal phenotype, we used a genetic approach to select suppressors allowing survival. This method, that we previously named FASTBAC-Seq for Functional AnalysiS of Toxin-antitoxin in BACteria by deep Sequencing, allows the mapping of intragenic toxicity suppressors within a given TA locus (*Masachis and Darfeuille, 2018*). Applying FASTBAC-Seq to the *aapA3*/IsoA3 TA locus revealed that single point mutations are sufficient to counteract the lethality caused by the absence of antitoxin. Unexpectedly, one-third of suppressors mapped to non-coding regions of the toxin mRNA. Some of them target well-known regulatory elements, such as the toxin promoter and the SD sequence. Remarkably, we show that one of the suppressors located in the SD sequence does not act at the sequence level but at the mRNA structural level. Indeed, this mutation inhibits translation of the active mRNA by stabilizing an RNA hairpin in which the SD sequence is masked by an upstream-encoded aSD sequence (aSD1). A synonymous mutation within the Open Reading Frame (ORF) acts similarly but on another hairpin in which the SD sequence is masked by a downstream-encoded aSD (aSD2). These suppressor mutations reveal two transient hairpin structures that sequentially form during transcription and which are then replaced by a more stable LDI upon transcription termination. Our results indicate that, in addition to the post-transcriptional control achieved via a stable LDI, metastable structures are also required to prevent premature toxin expression in a co-transcriptional manner.

## Results

### The small antisense RNA IsoA3 is essential to prevent AapA3 translation

We previously studied the regulation of *aapA1*/IsoA1, a member of the *aapA*/IsoA type I TA family recently identified in *H. pylori* (*Arnion et al., 2017*). Here, we studied the *aapA3*/IsoA3 module (*Figure 1A*; for sequence details see *Figure 1—figure supplement 2*). Like other TA systems of this family, the *aapA3*/IsoA3 locus codes for an antisense RNA, IsoA3 (80 nucleotides), encoded on the opposite strand of a small ORF, AapA3. The AapA3 peptide (30 amino acids) shares 60% sequence identity with the AapA1 peptide, whose ectopic expression is toxic in *H. pylori* (*Arnion et al., 2017*). Here, we first investigated whether *aapA3* expression from the chromosome is toxic. For this purpose, we inactivated the antitoxin promoter by introducing two point mutations in its −10 box, while

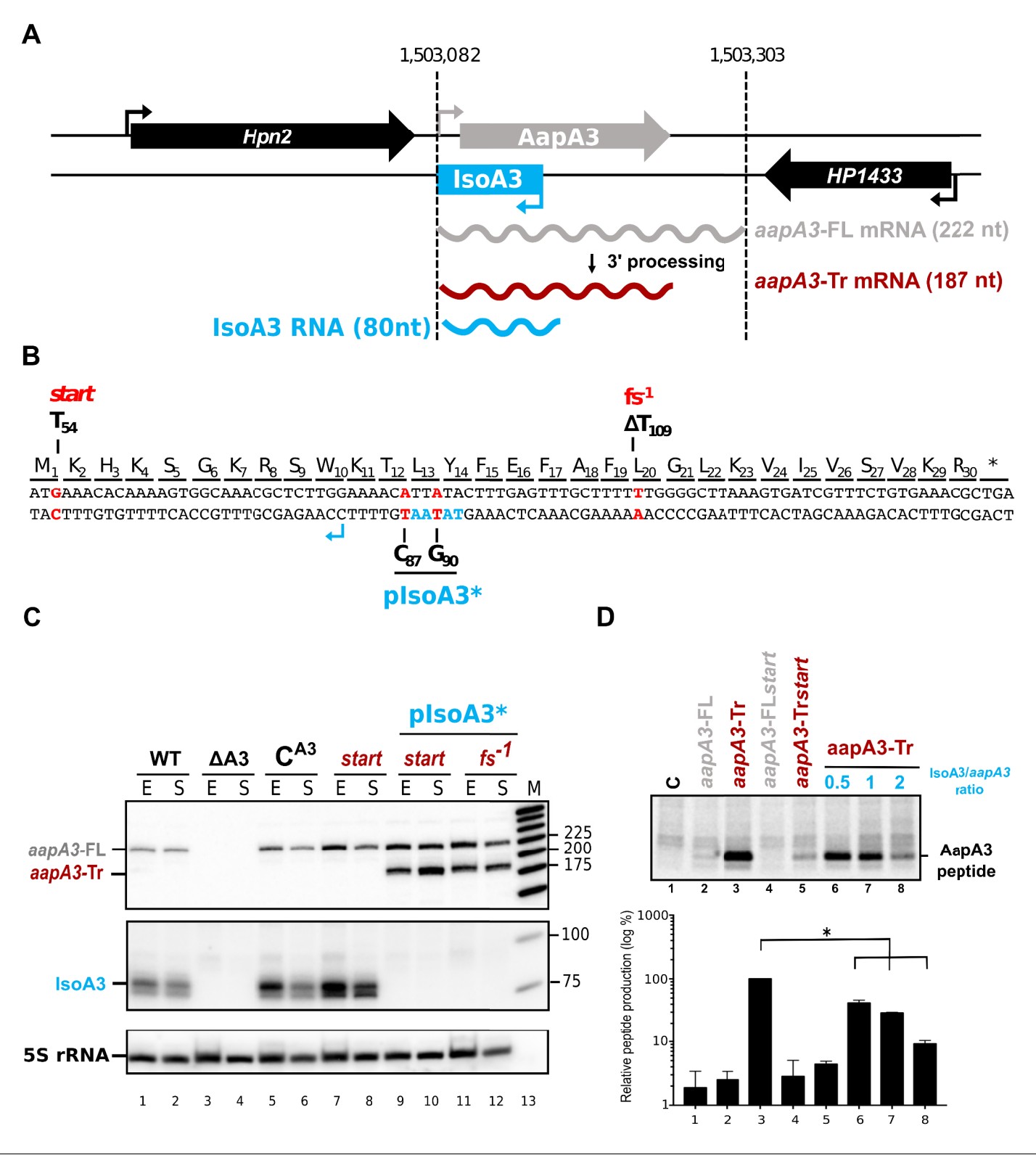

**Figure 1.** IsoA3 small RNA is essential to prevent AapA3 translation. (**A**) Organization of the aapA3/IsoA3 locus in the H. pylori 26695 strain. Grey arrow, AapA3 ORF; blue box, IsoA3 RNA; small bent arrows, -10 box of each transcript. Grey, red and blue wavy lines represent aapA3-FL (full- length), aapA3-Tr (3'-end truncated) and IsoA3 transcripts, respectively. Their approximate length is also indicated. Detailed locus coordinates can be found at the T1TA database website under the following link: https://d-lab.arna.cnrs.fr/display/display_feature_table/TA00126). For details about the aapA3/IsoA3

*Figure 1 continued on next page*

*Figure 1 continued*

locus deletion strategy see *Figure 1—figure supplement 2*. (B) Nucleotide and amino acid sequence of AapA3 ORF with hallmarks. The sequence of the IsoA3 promoter (-10 box) is shown in blue. The nucleotides that were mutated to inactivate the IsoA3 promoter (pIsoA3*) and the AapA3 start codon (start), and to create a -1 frameshift (fs -1 ) are shown in red. (C) The 'WT' strain corresponds to the 26695 H. pylori strain containing the K43R mutation in the rpsL gene, which confers resistance to streptomycin. The ΔA3 strain is the parental strain in which the aapA3/IsoA3 locus has been replaced by the rpsL Cj -erm cassette (ΔaapA3/IsoA3::rpsL Cj -erm). The C A3 and start strains correspond to the ΔA3 strain complemented with the WT aapA3/IsoA3 locus and with the locus mutated at the start codon (G54T), respectively. The two strains inactivated for the IsoA3 promoter (pIsoA3*) also contain a frameshift mutation (fs -1 ) or a mutation in the start codon (start). Total RNA from stationary (S) or exponential (E) growth phase of the indicated strains was isolated and subjected to Northern blot. The aapA3-FL, aapA3-Tr, and IsoA3 transcripts are shown. 5S rRNA assessed proper loading. (D) Translation assays were performed with 0.5 μg of aapA3 mRNAs in absence or presence of IsoA3, in 0.5, 1 or 2 molar ratios. [ 35 S]-Met was used for labeling. The control lane (C) shows the translation background obtained without exogenous mRNA.

DOI: https://doi.org/10.7554/eLife.47549.002

The following figure supplements are available for figure 1:

**Figure supplement 1.** General model of the *aapA*/IsoA TA systems regulation in *Helicobacter pylori*.

DOI: https://doi.org/10.7554/eLife.47549.003

**Figure supplement 2.** Details on *aapA3*/IsoA3 locus deletion and deep-sequencing approaches.

DOI: https://doi.org/10.7554/eLife.47549.004

maintaining the amino acid sequence of the toxin (*Figure 1B*, pIsoA3* in all figures), as previously described for the IsoA1 promoter (*Arnion et al., 2017*). Insertion of these mutations on the chromosome was performed using a counter-selection cassette, which allows the generation of unmarked transformants (*Dailidiene H. al., 2006*). Briefly, the TA locus of a streptomycin-resistant *H. pylori* 26695 strain (K43R) was replaced by the *rpsL$_{Cj}$-erm* double-marker cassette, giving rise to the streptomycin-sensitive Δ*aapA3*/IsoA3::*rpsL$_{Cj}$-erm*/K43R strain (*Masachis and Darfeuille, 2018*). Then, we performed gene replacement assays using PCR constructs carrying either a wild-type (WT) or an antitoxin-inactivated (pIsoA3*) TA locus. Strains that had undergone homologous recombination were selected on streptomycin. However, no transformants were obtained unless a non-sense or frameshift mutation was introduced in the *aapA3* ORF. This result indicated that, in the absence of IsoA3 synthesis, the AapA3 toxin expression from its chromosomal locus is constitutive and too toxic to obtain a viable strain.

Total RNA of two strains carrying the mutations inactivating the Iso3 promoter and either a mutation at the start codon (*start*) or a −1 frameshift mutation leading to a premature stop codon (*fs$^{-1}$*) (*Figure 1B*) were analyzed by Northern Blot (*Figure 1C*). The absence of IsoA3 transcript (*Figure 1C*, lanes 9–12) confirmed the successful inactivation of the IsoA3 promoter. As a control, the complementation of the Δ*aapA3*/IsoA3::*rpsL$_{Cj}$-erm*/K43R strain (ΔA3) with the WT *aapA3*/IsoA3 locus (C$^{A3}$) was successfully achieved with no change in the expression pattern (*Figure 1C*, compare lanes 5–6 with lanes 1–2). Two *aapA3* mRNA species were detected in absence of IsoA3 expression (*Figure 1C*, lanes 9 to 12): a long transcript of 225 nt, which was denoted *aapA3*-FL (full-length) and a shorter transcript lacking the last 35 nt (*aapA3*-Tr). This latter was not detected in presence of IsoA3 RNA (*Figure 1C*, lanes 1, 2, 5–8) and corresponds to the truncated mRNA species previously described for the *aapA1*/IsoA1 homolog (*Arnion et al., 2017*). Translation assays demonstrated that only the truncated mRNA is efficiently translated in vitro (*Figure 1D*, lane 3). Translation of *aapA3*-FL (*Figure 1D*, lane 2) or of *aapA3*-FL containing a non-sense mutation in the start codon (*Figure 1D*, lane 4) was close to the translational background obtained in absence of exogenous mRNA (*Figure 1D*, lane 1). Importantly, Northern blot analysis revealed that the absence of IsoA3 leads to the accumulation of *aapA3*-Tr without affecting the amount of *aapA3*-FL (*Figure 1C* lanes 9–12), indicating that IsoA3 specifically targets *aapA3*-Tr in vivo. In vitro structure probing of the two AapA3 mRNA species further confirmed that IsoA3 only interacts with the *aapA3*-Tr mRNA (*Figure 2*, compare lanes 4–7 with 10–13). Base-pairing between both transcripts creates an extended RNA heteroduplex of 80 base-pairs (*Figure 2*, lane 4) that is translationally inert, as shown by in vitro translation assays (*Figure 1D*, lanes 6–8). Remarkably, none of the IsoA RNAs produced from the five other *aapA*/IsoA chromosomal loci (I, II, IV, V and VI) can replace the absence of IsoA3 expression demonstrating that their regulation is strictly module-specific, as previously suggested by in vitro translation assays (*Sharma et al., 2010*).

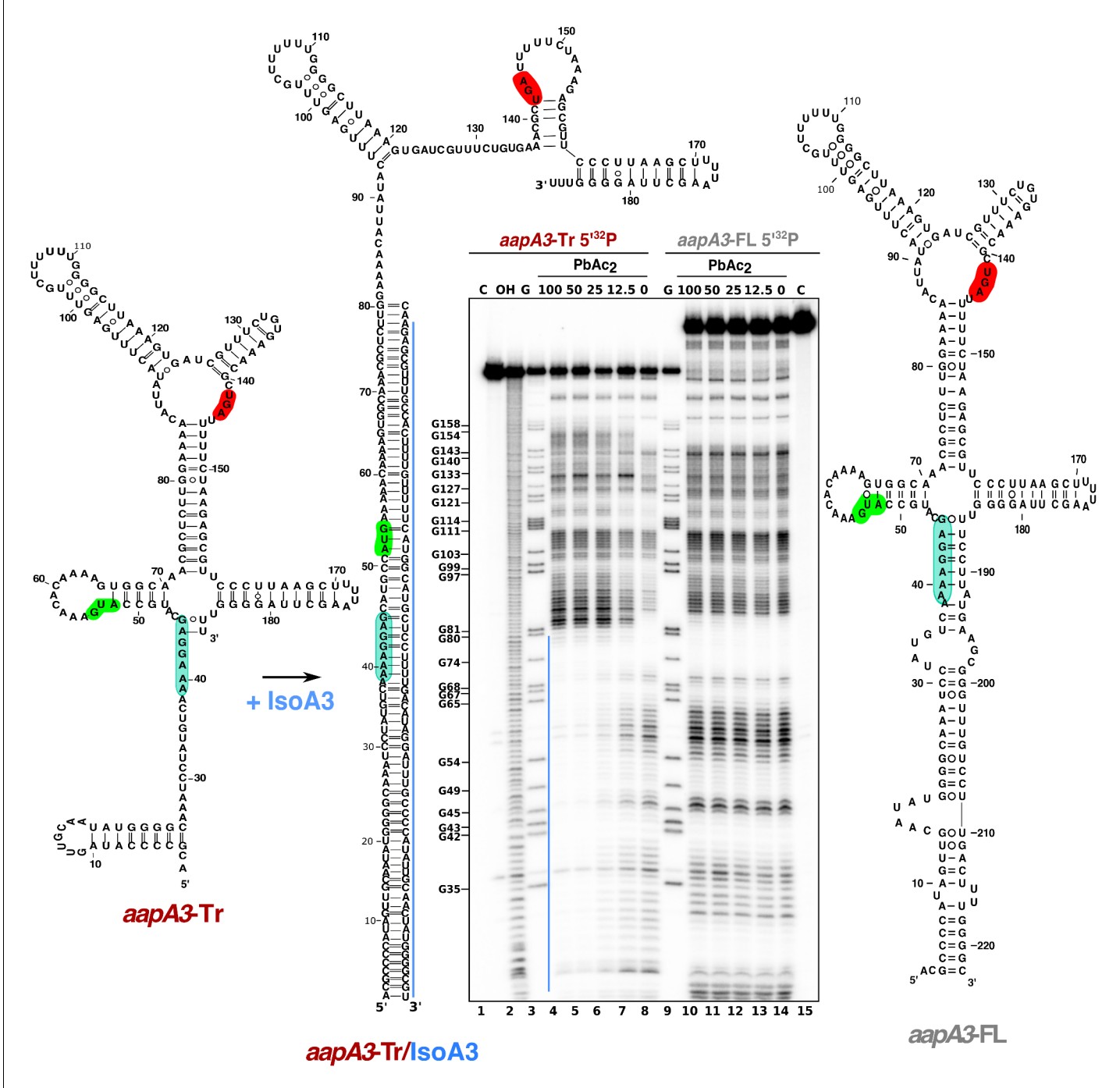

**Figure 2.** IsoA3 inhibits *aapA3*-Tr translation by masking its SD region. ~0.1 pmol of 5'-end [³²P]-labeled in vitro transcribed *aapA3*-FL and *aapA3*-Tr RNAs were subjected to lead probing in presence of increasing concentrations of IsoA3 (0–100 nM). Untreated RNA (lanes 1 and 15, denoted C) and partially alkali-digested RNA (denoted OH, lane 2) served as control and ladder, respectively. Positions of all G residues revealed upon T1 digestion under denaturing conditions (lanes 'G') are indicated relative to the transcription start site of the *aapA3* gene. Cleaved fragments were analyzed on an 8% denaturing polyacrylamide (PAA) gel. 2D structure predictions were generated with the RNAfold Web Server (*Gruber et al., 2008*) and by introducing additional constraints based on in vitro structure probing. The VARNA applet (*Darty et al., 2009*) was used to draw the diagrams. The region involved in duplex formation between IsoA3 and *aapA3*-Tr mRNA is indicated with a blue line; the start codon, stop codon and SD sequence of AapA3 are highlighted in green, red and turquoise, respectively.

DOI: https://doi.org/10.7554/eLife.47549.005

Altogether, these results show that IsoA3 represses *aapA3* constitutive expression at the translational level by forming a stable RNA heteroduplex.

## Decoding AapA3 toxicity determinants with nucleotide resolution

To identify the toxicity determinants of this TA locus, we next took advantage of the lethality induced by the chromosomal inactivation of the IsoA3 antitoxin to search for toxicity suppressors. To this end, we performed the same gene replacement assays as described above, using two PCR constructs carrying either a WT TA locus or two synonymous mutations inactivating the IsoA3 promoter (pIsoA3*). Two additional PCR fragments containing either a point mutation inactivating the start codon (*start*) or both mutations (*start*/pIsoA3*) were also used as controls (*Figure 3A*). Each PCR construct was transformed into the Δ*aapA3*/IsoA3::*rpsL*$_{Cj}$-*erm*/K43R *H. pylori* strain and Str$^R$ transformants were selected on streptomycin-containing plates. For each transformation, the number of streptomycin-resistant colonies was determined and normalized to the total number of transformed cells (*Figure 3B*). As expected, inactivation of IsoA3 promoter led to a strong reduction (1.83 log-fold) in the fraction of Str$^R$ cells compared to that obtained with the WT or double-mutant *start*/pIsoA3* constructs (*Figure 3B*). These results confirmed that in the absence of IsoA3, the chromosomal expression of AapA3 is highly toxic. Notably, the number of streptomycin-resistant colonies obtained with pIsoA3* was slightly higher than that obtained when no DNA was used (*Figure 3B*). The sequencing analysis revealed that the Str$^R$ cells obtained in absence of DNA were all phenotypic revertants having mutated the *rpsL*$_{Cj}$ gene (*Masachis et al., 2018*). In contrast, sequencing of the TA locus of approximately hundred pIsoA3* transformants revealed that all of them contained point mutations in the AapA3 ORF (*Masachis et al., 2018*). Thus, our genetic approach selected mutations that suppress toxicity allowing the generation of recombinant strains lacking antitoxin expression.

To explore the complete landscape of suppressors, we next scaled-up the transformation assay using the WT or pIsoA3* PCR products as DNA substrates. This approach, called FASTBAC-Seq, has been described recently (*Masachis et al., 2018*). Consistent with the above-mentioned first transformation assay, deep-sequencing data analysis showed that 97.7% of the pIsoA3* transformants contained mutations (*Figure 3C*). A strong mutation rate (51.2%) was also observed with the WT PCR product (*Figure 3C*), which results from an unanticipated technical artifact linked to PCR amplification (see Material and methods section). Analysis of the number of mutations per read in the complete sequencing dataset (all replicates combined) showed that more than half of the pIsoA3* transformed clones (51.8% out of ~5.1 million) were mutated at a single nucleotide position (*Figure 3C*). This result demonstrates that single point mutations are sufficient to abolish AapA3 toxin activity and/or expression. A low number of pIsoA3* transformants (2.3%) had a WT locus sequence (*Figure 3C*, pIsoA3* zero mutation), which can be explained by the sequencing error rate (around 1.5%) and/or suppressor mutations lying in regions outside the TA locus (i.e. outside the amplicon). Single-nucleotide mutations were mainly substitutions, which are favored by PCR biases. Only 4% were insertions and deletions (indels). Hierarchical clustering analysis revealed that the location and frequency of single substitutions were highly similar in the three biological replicates, indicating that the locus coverage was close to optimum (*Figure 3—figure supplement 1*). Contrary to substitutions (which were found in both coding and non-coding regions), the single-nucleotide indels were almost exclusively present in the AapA3 ORF, generating truncated or longer forms of the peptide. Moreover, in some cases, they were not present in all three replicates, reflecting their underrepresentation in the PCR products (*Figure 3—figure supplement 1*). Expectedly, the highest mutation densities (defined as the number of mutated nucleotides divided by the total number of nucleotides in the region of interest) were observed in the AapA3 ORF (53%), and in well-known regulatory regions such as the −10 box of the toxin mRNA (66.7%, *Figure 3—figure supplement 1*) and the SD sequence (42.8%) (*Figure 3D*). For the *aapA3* −10 box, out of the six positions (5'-TAGGAT-3'), suppressor mutations were mostly found in the first two and last nucleotides (*Figure 3—figure supplement 2*). This result allowed us to determine the minimal functional *aapA3* −10 box motif 5'-TANNNT-3', which is in perfect agreement with the previously determined consensus sequence (*Sharma et al., 2010*). This result also confirmed that the arbitrarily-chosen False Discovery Rate cutoff (padj ≤0.05) was stringent enough to avoid false positives but permissive enough to allow the identification of suppressor substitutions. Remarkably, 17 mutations were unveiled in the 5' and 3' untranslated regions (UTR; *Figure 3D*). Several of these point mutations created mismatch pairing,

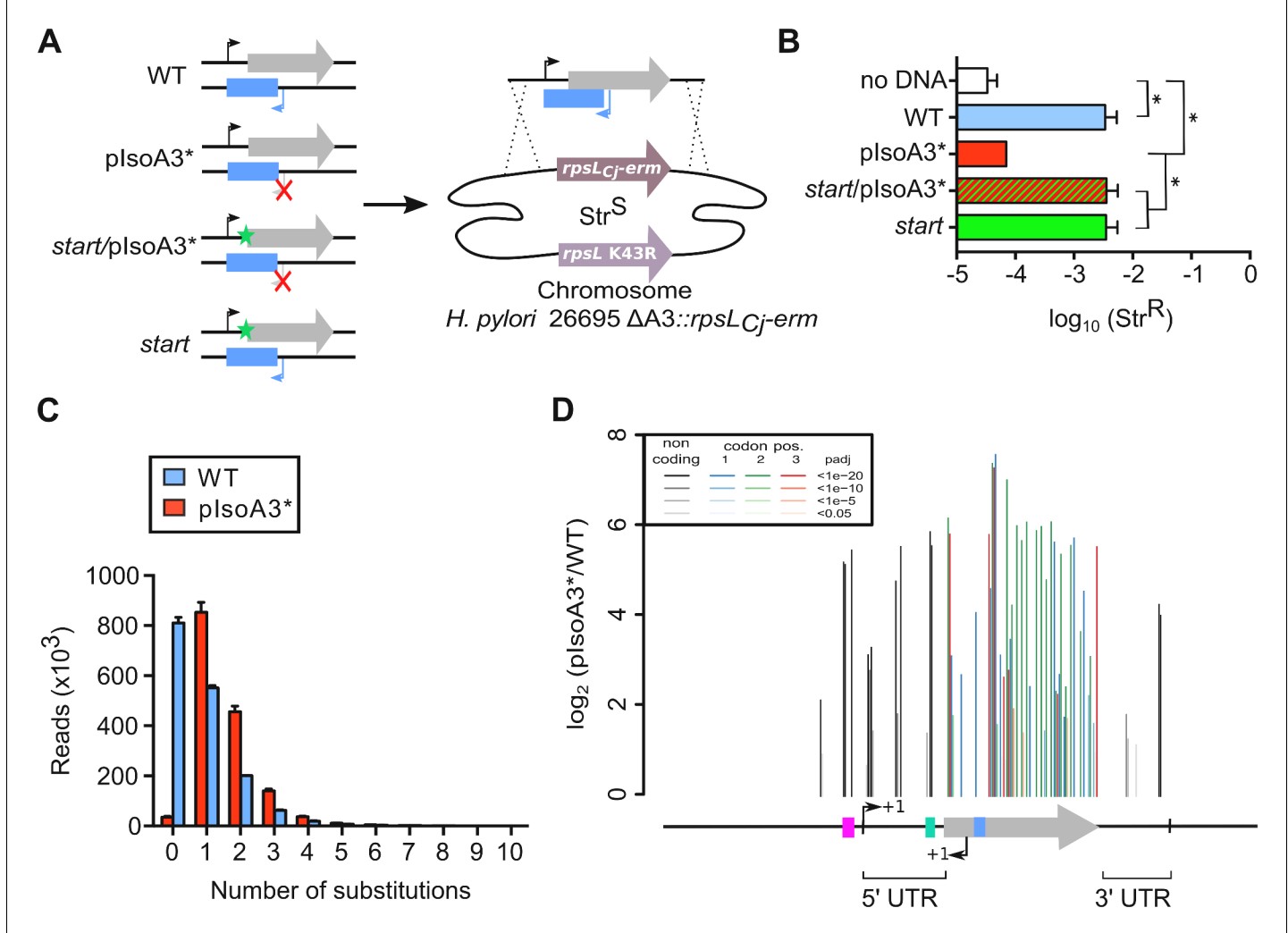

**Figure 3.** Unveiling intragenic toxicity determinants with nucleotide resolution. (**A**) PCR fragments used for transformation of the ΔA3 strain (Δ*aapA3*/ IsoA3::*rpsL_Cj-erm)* are shown. A green star indicates a mutation in the start codon (G54T). The red cross indicates the two mutations (A87G and A90C) introduced to inactivate the IsoA3 promoter (pIsoA3*). Gene replacement at the locus was selected by plating transformants on streptomycin. (**B**) Transformation efficiency (number of Str[R] transformants divided by the total number of transformed cells) was determined for each construct. A control in which the PCR fragment was replaced by H2O (no DNA) is also shown. Error bars represent standard deviations (s.d); *n = 3* biological replicates (*p<0.05; values according to unpaired *t*-test). (**C**) Number of reads containing 0 to 10 substitutions in the sequenced amplicon of 426 nt encompassing the *aapA3*/IsoA3 TA locus. Error bars represent s.d; *n = 3* biological replicates. (**D**) Positional analysis of single-nucleotide substitutions on the *aapA3*/ IsoA3 locus. Bar plot shows the log2 fold-change (pIsoA3*/WT ratio) for the 70 positions with an adjusted p-value (padj) lower than 0.05. Bars are drawn with different shades of gray according to the p-value in the non-coding regions, and with different shades of color within the coding region (red, green, or blue, for the first, second, and third codon position, respectively). The relevant sequence elements are indicated by arrows and boxes under the graph. 5' UTR, 5' untranslated region; purple box, *aapA3* −10 box; small bent arrows, +1 transcription start site (TSS) of *aapA3* and IsoA3; turquoise box, *aapA3* SD sequence; large gray arrow, *aapA3* ORF; small blue box, IsoA3 −10 box; 3' UTR, 3' untranslated region. A comparison of the distribution and relative frequency of single-nucleotide suppressors in the strains transformed with the WT or pIsoA3* PCR constructs can be found in *Figure 3—figure supplement 1*.

DOI: https://doi.org/10.7554/eLife.47549.006

The following source data and figure supplements are available for figure 3:

**Source data 1.** Raw data to determine the transformation efficiency in *Figure 3B*.

DOI: https://doi.org/10.7554/eLife.47549.010

**Figure supplement 1.** Comparison of the distribution and relative frequency of single-nucleotide suppressors in the WT and pIsoA3* *aapA3*/IsoA3 modules.

DOI: https://doi.org/10.7554/eLife.47549.007

**Figure supplement 2.** Defining and validating the *aapA3* promoter with nucleotide resolution.

*Figure 3 continued on next page*

*Figure 3 continued*

DOI: https://doi.org/10.7554/eLife.47549.008

**Figure supplement 3.** Single-nucleotide suppressor mutations located in the 5' and 3' terminal stem-loops of the active *aapA3* mRNA.

DOI: https://doi.org/10.7554/eLife.47549.009

probably destabilizing two terminal stem-loop located at both ends of the active mRNA (*aapA3*-Tr) (*Figure 3—figure supplement 3*). In the present work, we have focused our study to mutations lying around the RBS.

## Two suppressor mutations in the 5' UTR impede AapA3 translation

The genetic selection of suppressors allowed us to determine the minimal functional sequence of the toxin SD sequence. Among the seven positions (5'-AAAGGAG-3'), substitutions at the two central guanine nucleotides (positions +42 and +43) were the most frequently mutated in absence of antitoxin (*Figure 4A*), and one can deduce the minimal SD sequence to be 5'-AGG-3' or 5'-GGA-3'. As expected from PCR biases (*Beaudry and Joyce, 1992*), although the transition G > A was preferentially enriched in both cases, the G > C and G > T transversion mutations were also selected. Strikingly, a less-frequent transversion mutation (A > T) was selected within the SD sequence at position +40 (A40T, *Figure 4A*). Moreover, another unique transversion upstream of the SD sequence was identified (A28C, *Figure 4A*). Strains containing these atypical mutations were constructed and further analyzed.

Due to their proximity to the SD sequence, we first tested whether these mutations could affect AapA3 translation in vivo. Due to the lack of AapA3-targeting antibodies, its translation was assessed indirectly by polysome fractionation coupled to Northern blot analysis. As a control, we used a suppressor mutation isolated during our FASTBAC-Seq selection, which suppresses the toxin activity but not its expression. We thus generated a strain containing the pIsoA3* mutations in combination with a mutation in the toxin ORF converting a phenylalanine at position 19 (of the peptide) into a serine (T107C, *Figure 4B*). Polysome fractionation of this strain confirmed that the toxin full-length mRNA (*aapA3*-FL) is mainly found in non-ribosomal fractions (only 3% present in polysomes, lanes 10 to 14 in *Figure 4B*; see *Figure 4—figure supplement 1* for quantification), whereas the truncated isoform (*aapA3*-Tr) is associated with the monosome and disome fractions (73% present in these fractions; *Figure 4B* and *Figure 4—figure supplement 1A*). The absence of the *aapA3*-Tr form in heavier polysomes is probably due to the short length of the ORF (90 nt), which cannot accommodate more than two ribosomes. This result clearly confirmed that the *aapA3*-FL is indeed a translationally inert isoform, whereas the *aapA3*-Tr is translationally active. Hence, polysome fractionation is a powerful tool to study translation efficiency of toxin mRNAs in vivo.

We then analyzed the efficiency of toxin mRNA translation in strains containing a mutation either in the start codon (*start*) or in the SD sequence (G43A) (*Figure 4B*). In both cases, the active mRNA isoform (*aapA3*-Tr) was nearly absent from the polysome fractions. Instead, significant levels of *aapA3*-Tr mRNA degradation products were detected in the top of the gradient, which may arise from the lack of ribosome protection and/or the extended time of sample collection and treatment prior to RNA extraction. The strong degradation observed for the *start* strain impeded the quantification of *aapA3*-Tr. For the G43A strain, the relative amount of *aapA3*-Tr mRNA in translating fractions was strongly reduced (approximately 4%, lanes 10 to 14, *Figure 4B* and *Figure 4—figure supplement 1E*) compared to the T107C strain. This result confirmed that the single G43A mutation was sufficient to prevent ribosome binding, consequently impairing translation of the toxin. Remarkably, a similar result was observed for the A28C and A40T suppressor mutations. The A40T mutation had the strongest effect with only 7% of *aapA3*-Tr associated with translating ribosomes, compared to 19% for the A28C strain (*Figure 4B* and *Figure 4—figure supplement 1C,D*). In vitro translation assays (*Figure 4—figure supplement 2C*) further confirmed that the A28C and A40T suppressor mutations, like the G43A mutant, act by reducing AapA3 translation efficiency.

Together, these results demonstrate that a single mutation within the 5' UTR, either inside or outside the SD sequence, is able to overcome antitoxin absence by impeding toxin translation.

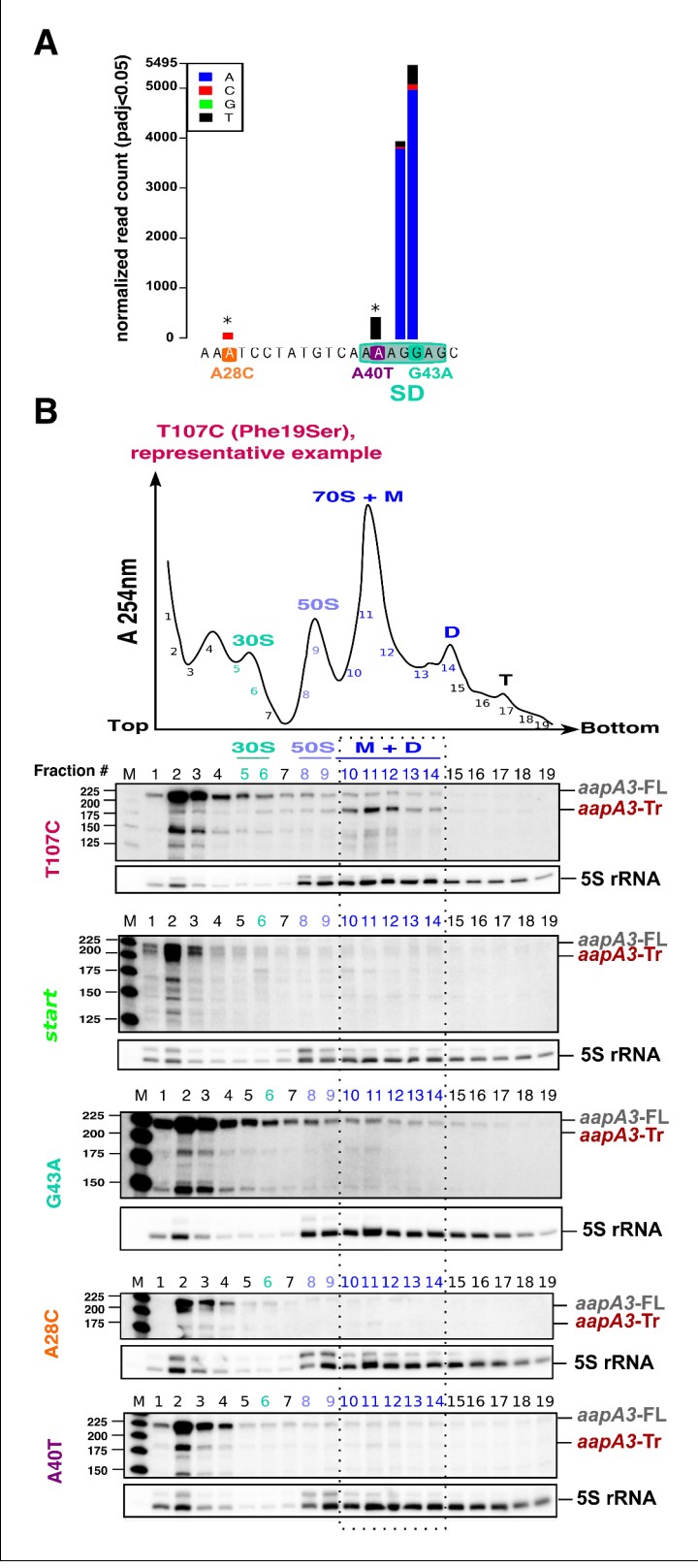

**Figure 4.** Single point suppressor mutations in the 5' UTR of the *aapA3* mRNA inhibit its translation. (**A**) Nucleotide substitutions within the *aapA3* 5' UTR that are significantly enriched (pajd ≤0.05, 'nucleotide-specific' analysis, see Materials and methods section) in pIsoA3* compared to WT. Asterisks above the bars indicate transversion mutations. The SD sequence is boxed. (**B**) Cell lysates of the indicated *aapA3* variant strains were

*Figure 4 continued on next page*

*Figure 4 continued*

subjected to ultracentrifugation through a sucrose gradient. A representative $A_{254nm}$ profile of the T107C strain is shown. Peaks of the free 30S and 50S subunits, 70S ribosomes (free ribosomes and monosomes (M)), and polysomes (D, disomes; T, trisomes) are indicated. RNA was extracted from each fraction and equal volumes of each fraction were subjected to Northern blot analysis. The different transcripts *aapA3*-FL, *aapA3*-Tr, and 5S rRNA (loading control) are indicated. The vertical dashed lines delineate the limits corresponding to 70S, monosome and disome fractions. Quantification of the relative band intensity of the *aapA3* mRNAs in each fraction can be found in *Figure 4—figure supplement 1*.

DOI: https://doi.org/10.7554/eLife.47549.011

The following source data and figure supplements are available for figure 4:

**Figure supplement 1.** Quantification of the relative *aapA3* mRNA band intensity from polysome fractionation Northern Blots shown in *Figure 4*.

DOI: https://doi.org/10.7554/eLife.47549.014

**Figure supplement 2.** The A28C, A40T and T78C suppressors inhibit *aapA3*-Tr mRNA translation.

DOI: https://doi.org/10.7554/eLife.47549.012

**Figure supplement 2—source data 1.** Numerical values of the 3 replicates of the in vitro translation assays shown in *Figure 4—figure supplement 2*.

DOI: https://doi.org/10.7554/eLife.47549.013

## The suppressor A28C and A40T mutations inhibit toxin translation by stabilizing a SD-sequestering hairpin

We next asked by which mechanism A28C and A40T mutations inhibit translation. Both substitutions lie in a single-stranded region upstream of the minimal SD sequence (5'-AGGA-3'), which may be crucial for translation initiation (*Figure 2*). However, in both cases, a unique type of transversion mutation was selected (*i.e.*, A28T and A40C were not selected) suggesting that the mutations may act at the structure rather than at the primary sequence level. Indeed, secondary structure predictions with the RNAfold algorithm (*Gruber et al., 2008*) suggested that both mutations could stabilize a local hairpin in which the SD is sequestered by an upstream aSD sequence (anti-SD sequence 1 [aSD1], *Figure 5A*). Indeed, while the A28C suppressor transversion extended this hairpin by one G-C base-pair, the A40T mutation created two additional A-U base-pairs. To investigate whether this stabilization was responsible for the translation inhibition effect, we tested whether the combination of A40T and antagonist A33T mutations (see *Figure 5A*), which is expected to destabilize the stem-loop, restored a toxic phenotype. Due to this potential toxicity, a strain containing an additional mutation in the AapA3 start codon was also generated (A33T/A40T/*start*, *Figure 5B*). Transformation assay was performed as previously described (*Figure 3A*). As expected, the suppressor A40T mutation was not toxic (*Figure 5B*). However, a log-fold reduction of ~2 in the number of Str$^R$ transformants was observed with the A33T/A40T construct (*Figure 5B*). This effect disappeared when the toxin start codon was mutated (A33T/A40T/*start*, *Figure 5B*) demonstrating that the toxicity comes from the AapA3 peptide synthesis. This approach could not be used to study the A28C suppressor mutation since the non-compensatory mutation would lie within the SD sequence. Therefore, we tested the SD accessibility for all mutants (A40T, A28C, A40T/A33T) in vitro by performing an RNase H/oligonucleotide assay (*Figure 5C*, and *Figure 5—figure supplement 1*). Compared to the WT and A33T/A40T mutant, a reduced oligonucleotide accessibility was observed in vitro for both A28C and A40T *aapA3*-Tr RNAs, demonstrating that both mutations inhibit toxin expression by reducing SD accessibility.

Altogether these results indicate that both suppressor mutations are preventing translation initiation by stabilizing the SD sequestration within a local RNA hairpin instead of acting at the sequence level.

## A second SD-sequestering hairpin is embedded within the *aapA3* ORF

A synonymous substitution (T78C) at amino acid position 9 (Ser), which converts a UCU to a UCC codon was also selected in our suppressor selection. The presence of such a mutation was intriguing as it is located 27 nt after the start codon and it does not affect the amino acid sequence. To understand its potential effect on toxin expression, we constructed a strain containing this mutation together with the pIsoA3* mutation. Northern blot analysis showed that the T78C strain contains

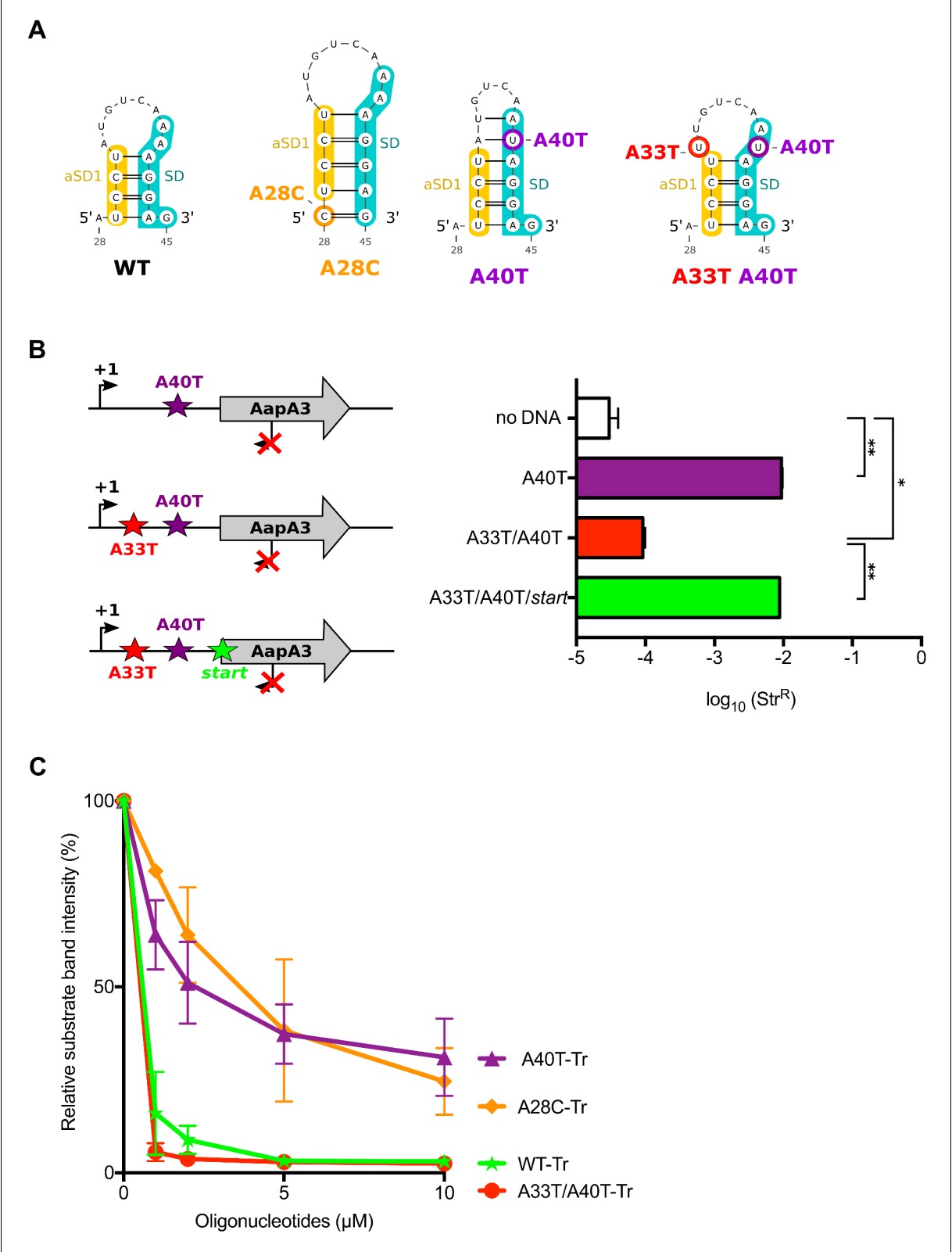

**Figure 5.** The A28C and A40T mutations suppress toxicity through SD sequestration. (**A**) Predicted hairpins involving the sequestration of the SD sequence by the aSD1 motif. Prediction of these secondary structures were carried out on the *aapA3*-Tr transcript (see *Figure 5—figure supplement 2* for visualizing the whole structure) using the same software as in *Figure 2*, but with no additional constraint. The A28C mutation (dark orange) generates an extra G-C base-pair; in the A40T mutant (purple), two additional A-U pairs stabilize the hairpin; the A40T antagonist mutation A33T is

*Figure 5 continued on next page*

*Figure 5 continued*

shown in red; the SD and aSD1 sequences are shown in turquoise and yellow, respectively. (B) (left) PCR constructs used to assess the SD sequestering structure by transformation assay. (right) For each transformation with the indicated PCR constructs, the number of Str[R] obtained per total number of transformed cells was calculated and plotted on a log scale. Error bars represent s.d; *n = 3* biological replicates. (\*\*\*p<0.0001; \*p=0.001 according to unpaired *t*-test). (C) Graph representing the RNase H assays. The position of the oligonucleotides (FA644 for WT and A40T; FA651 for A33T/A40T; and FA652 for A28C, see *Table 1*) used in the RNase H protection assay is indicated by a black arrow along the first 45 nucleotides of the *aapA3* mRNA (*Figure 5—figure supplement 2*). 30 fmol of internally labeled WT and mutated *aapA3*-Tr transcripts were incubated with 0 to 100 pmoles of each specific DNA oligonucleotide and subjected to digestion by *E. coli* RNase H1. Digestion products were analyzed on an 8% PAA denaturing gel (*Figure 5—figure supplement 1*). Substrate consumption was quantified as relative substrate band intensity, 100% corresponding to the intensity obtained in absence of oligonucleotide. Error bars represent the s.d; *n = 2* technical replicates.

DOI: https://doi.org/10.7554/eLife.47549.015

The following source data and figure supplements are available for figure 5:

**Source data 1.** Raw data to determine the transformation efficiency in *Figure 5B*.
DOI: https://doi.org/10.7554/eLife.47549.018
**Source data 2.** Numerical values of the graph shown in in *Figure 5C*.
DOI: https://doi.org/10.7554/eLife.47549.019
**Figure supplement 1.** Gel analysis of RNase H/oligonucleotide accessibility assays.
DOI: https://doi.org/10.7554/eLife.47549.017
**Figure supplement 2.** Location of the mutations working through the anti-SD sequence 1 (aSD1) in the *aapA3*-Tr mRNA.
DOI: https://doi.org/10.7554/eLife.47549.016

similar amounts of both *aapA3*-FL and *aapA3*-Tr mRNA isoforms than the other A28C and A40T suppressor strains (*Figure 4—figure supplement 2B*). We next tested the translatability of AapA3 in vivo by performing polysome fractionation coupled to Northern blot analysis. The percentage of *aapA3*-Tr found in the monosome and disome fractions of the T78C strain (*Figure 6A* and *Figure 4—figure supplement 1B*) was much lower than that observed for the control T107C strain (34% vs 73%), but significantly higher than the one observed for the A28C and A40T suppressors (19% and 7%, respectively; *Figure 4B*). In vitro translation assays confirmed these results (*Figure 4—figure supplement 2*), demonstrating that the T78C suppressor acts by inhibiting AapA3 translation.

Secondary structure prediction revealed another putative SD-sequestering hairpin involving an aSD sequence (aSD2) embedded within the AapA3 ORF (*Figure 6B*). This hairpin is mutually exclusive from the aSD1-containing hairpin. As for the A28C and A40T suppressor mutations, the T78C transition was expected to stabilize the hairpin by replacing a G-U by a G-C pair. To address the accessibility of this region, an RNase H protection assay was performed, using the FA633 oligonucleotide (*Figure 6C* and *Figure 6—figure supplement 1*). Remarkably, a strong reduction in SD accessibility was observed in vitro for the T78C RNA compared to the WT (*Figure 6C and D*). Thus, a single hydrogen bond is sufficient to stabilize the sequestration of the SD sequence and to suppress toxicity. Importantly, despite being located within the AapA3 coding region, the T78C suppressor acts at the mRNA folding level.

Sequence conservation analysis of the AapA3 coding region in 49 *H. pylori* strains (*Figure 6—figure supplement 2*) revealed that the serine codon at position nine is one of the most highly conserved codons of the ORF, indicating a crucial role of this sequence, likely in the sequestration of the SD sequence. Only the UM066 strain (highlighted in pink in *Figure 6—figure supplement 2*) possesses a proline at this position, which probably abolishes peptide toxicity by disrupting the alpha-helix structure of the toxin (*Masachis et al., 2018*).

## Working model

We have shown that mutations in aSD1 and aSD2 sequences suppress toxicity probably by stabilizing two mutually exclusive short hairpins that inhibit toxin translation (*Figure 7*). These stabilized hairpins are formed in the *aapA3*-Tr but not in the *aapA3*-FL RNA (*Figure 7—figure supplement 1*), explaining why they inhibit translation without affecting the stability of the full-length form. Interestingly, our FASTBAC-Seq approach revealed that, out of thirteen possible aSD sequences present in the *aapA3* mRNA (*Figure 7—figure supplement 1*), only two (the two closest to the SD) could be mutated to suppress toxicity. Our results suggest that, in the WT context, these two hairpins are not stable enough and are exclusively formed transiently during transcription. We propose that they act

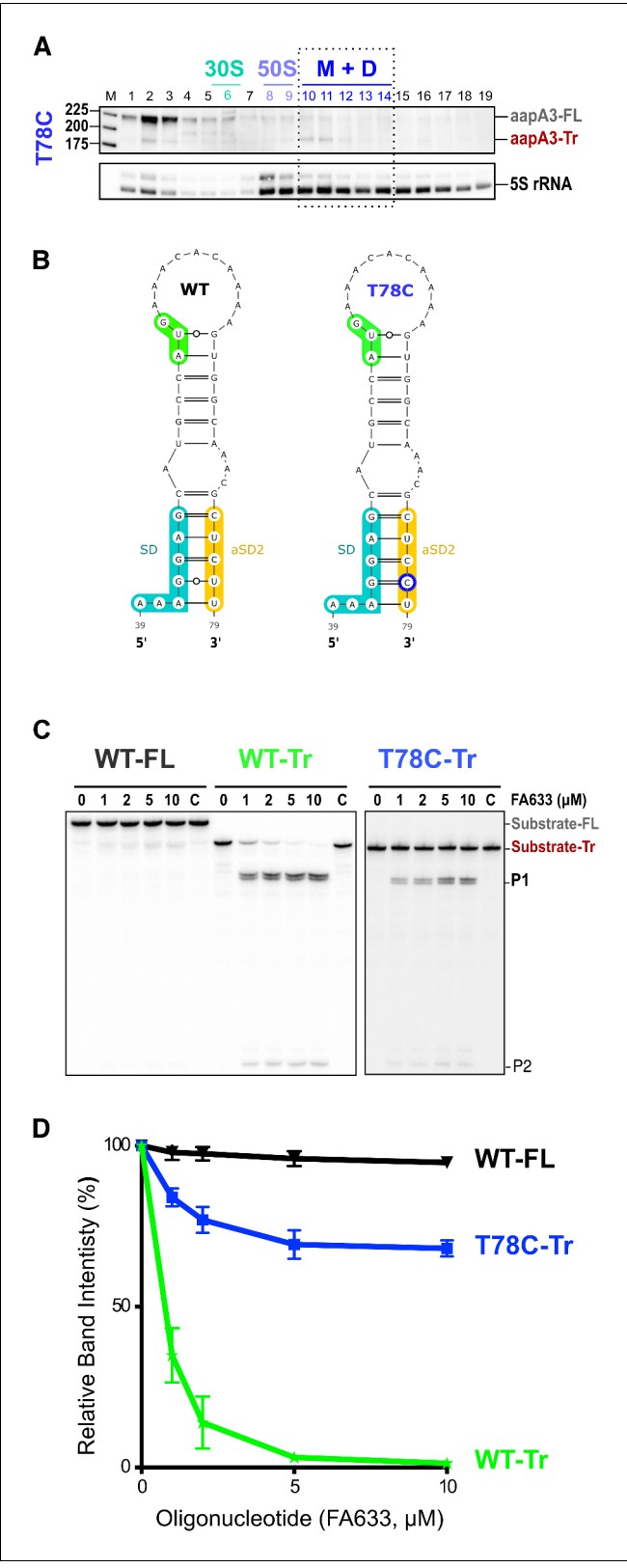

**Figure 6.** A synonymous mutation located within the toxin ORF inhibits *aapA3* mRNA translation via SD sequestration. (**A**) The cell lysate of the T78C strain was subjected to ultracentrifugation through a sucrose gradient. RNA was analyzed as in *Figure 4*. The different transcripts *aapA3*-FL, *aapA3*-Tr, and 5S rRNA (loading control) are indicated. M+D, monosomes + disomes. (**B**) Prediction of the secondary structure involving the

*Figure 6 continued on next page*

*Figure 6 continued*

second aSD sequence (aSD2). Prediction was carried out on the biologically relevant mRNA (*aapA3*-Tr) (see *Figure 6—figure supplement 1*). No additional constrains were used. The location of the oligonucleotide used in this assay (FA633) is shown by a black arrow in *Figure 6—figure supplement 1*. The T78C mutation is shown in dark blue, SD sequence in turquoise, anti-SD sequence in yellow and start codon in green. (C) A typical RNase H protection assay is shown. A total of 30 fmol of internally labeled *aapA3*-FL and *aapA3*-Tr RNA (WT or T78C) were incubated in presence of 0 to 100 pmoles of DNA oligonucleotide (FA633) and subjected to digestion by *E. coli* RNase H1. Lane C contains only the labeled substrate in absence of the enzyme. Two digestion products, P1 and P2, are indicated. (D) Substrate consumption was quantified as the relative substrate band intensity and plotted as a function of DNA oligonucleotide concentration. Error bars represent the s.d; *n* = 2 technical replicates.

DOI: https://doi.org/10.7554/eLife.47549.021

The following source data and figure supplements are available for figure 6:

**Source data 1.** Numerical values of the graph shown in in *Figure 6D*.

DOI: https://doi.org/10.7554/eLife.47549.024

**Figure supplement 1.** Location of the T78C suppressor mutation working through the anti-SD sequence 2 (aSD2) in the *aapA3*-Tr mRNA.

DOI: https://doi.org/10.7554/eLife.47549.022

**Figure supplement 2.** Nucleotide alignment of AapA3 coding region of 49 *Helicobacter pylori* strains.

DOI: https://doi.org/10.7554/eLife.47549.023

---

as functional metastable structures (MeSt1 and MeSt2, *Figure 7*), that is they form co-transcriptionally and sequentially to prevent premature toxin expression before a third aSD (aSD3) traps the *aapA3*-FL mRNA into a highly stable and translationally inert conformation (*Figure 7*). To confirm the existence of these transient RNA structures, we performed in vivo DMS RNA footprinting on strains expressing truncated versions of the aapA3 mRNA, which are expected to mimic two different nascent transcripts. The pattern of methylation confirmed that both RNA hairpins, MeSt1 and MeSt2, are formed in vivo (*Figure 8*). Moreover, this experiment also confirmed that the A40T mutation stabilizes the MeSt1 hairpin. Finally, the existence of these structures is well supported by a co-variation analysis showing the occurrence of several compensatory mutations in these hairpins in numerous *H. pylori* strains (*Figure 8—figure supplements 1* and *2*).

## Discussion

How bacteria modulate gene expression via RNA structure has been a fascinating topic for the last 30 years. This regulation is often achieved at the translation initiation step through the sequestration of the SD sequence in stable RNA hairpins that prevent ribosome binding to the RBS of the mRNA (*Duval et al., 2013*; *Duval et al., 2015*; *Meyer, 2017b*). Although recent advances using in vivo probing at the genome scale have confirmed that translation efficiency strongly correlates with the mRNA structure around the RBS (*Mustoe et al., 2018*), little is known about the influence of co-transcriptional folding on translation. Bacteria could, in principle, reduce or delay the translation of a specific mRNA by playing with its secondary structure while the mRNA is being made (*Lai et al., 2013*; *Zhu and Meyer, 2015*). In this article, we identified two functional RNA hairpins within a type I toxin-encoding mRNA for which a tight control of translation is essential. We propose that these hairpins correspond to metastable structures that form sequentially and transiently to occlude SD accessibility during mRNA synthesis.

### FASTBAC-Seq uses toxin lethality to identify suppressor mutations

To date, most studies on TA systems, including our previous work (*Arnion et al., 2017*), used artificial expression systems to characterize the effects of toxin expression. The use of such overexpression vectors is often a source of misinterpretations, as toxic proteins may not be found at such high concentrations under physiological conditions. To study the AapA3 toxin expression at the chromosomal level, we inactivated the endogenous IsoA3 antitoxin promoter as previously described for IsoA1 (*Arnion et al., 2017*). However, we were unable to obtain a viable strain without introducing additional mutations in the toxin-encoding gene. Suppressor mutations have been also reported in *B. subtilis* for two chromosomally encoded type I toxins with killer activity (*txpA*/RatA

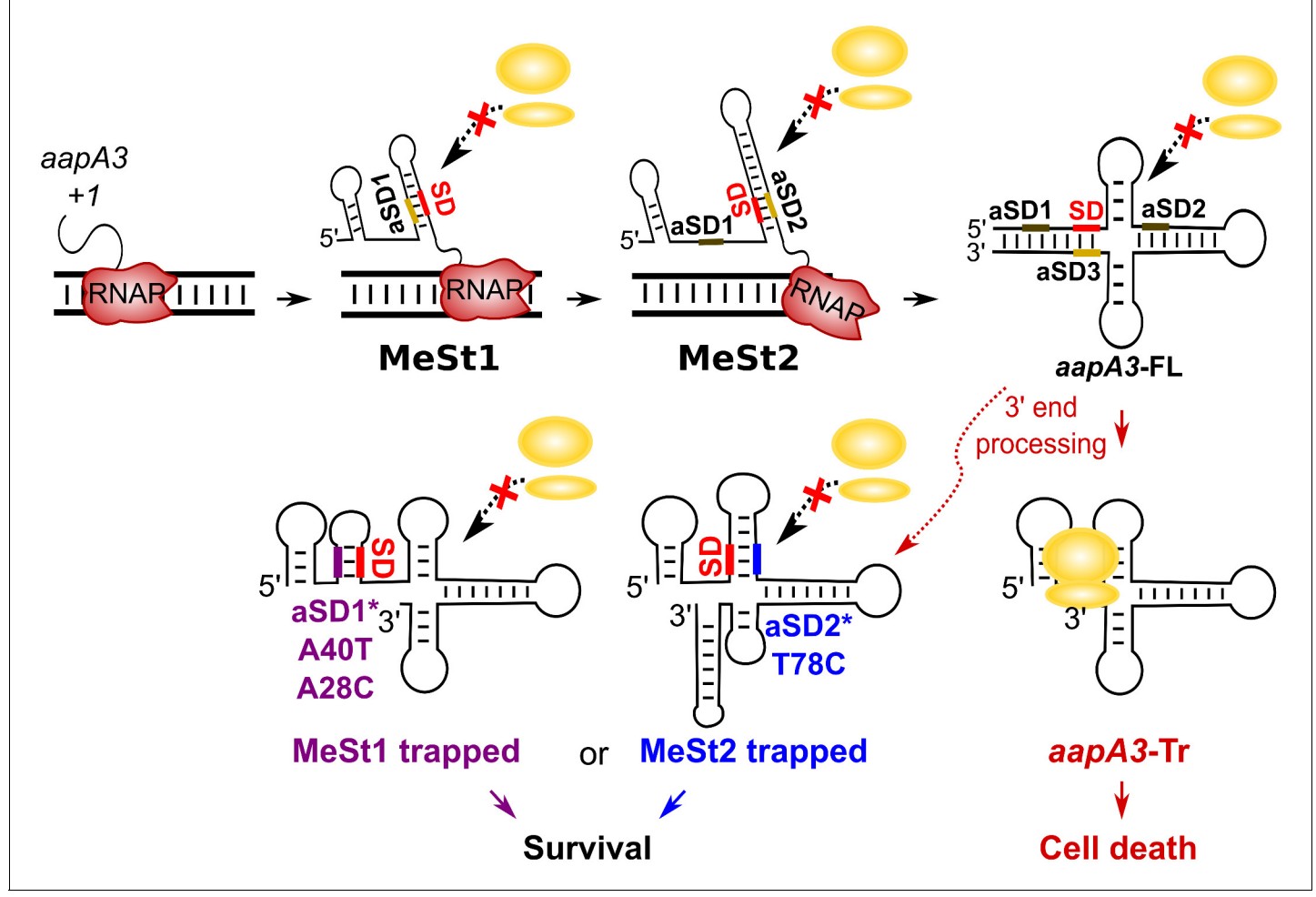

**Figure 7.** Working model of the *aapA3* co- and post-transcriptional regulation. Co-transcriptional folding of the *aapA3* mRNA leads to the generation of two successive SD-sequestering hairpins, which constitute metastable structures (MeSt) temporarily impeding ribosome access during transcription. The RNA polymerase is shown in red. Upon transcription termination, the *aapA3* full-length transcript (*aapA3*-FL) folds into a translationally inert conformation involving a 5'—3'-end long-distance interaction (LDI) in which the SD is sequestered by the aSD3 motif. A 3'-end nucleolytic truncation leads to the formation of *aapA3*-Tr, which is translationally active. In absence of IsoA3, its translation leads to cell death. We showed that the suppressor mutations A40T, A28C (purple) and T78C (blue) stabilize the metastable hairpins in the truncated isoform (involving the aSD1 and aSD2 sequences, respectively), leading to inhibition of *aapA3*-Tr translation. The suppressor strains can thus survive even in absence of IsoA3. The two successive MeSt structures act as thermodynamic traps to freeze the mRNA into translationally inert conformations.

DOI: https://doi.org/10.7554/eLife.47549.025

The following figure supplement is available for figure 7:

**Figure supplement 1.** Only three out of the 13 potential aSD sequences embedded in *aapA3* mRNA are functional.

DOI: https://doi.org/10.7554/eLife.47549.027

[*Silvaggi et al., 2005*] and *bsrG*/SR4 [*Jahn et al., 2012*]). Strikingly, lethality at the chromosomal level has not been reported for chromosomally-encoded TA loci in Gram-negative bacteria. Indeed, the killer activity observed for the plasmid-encoded *hok*/Sok TA system was even believed to be not conserved for the chromosomally-encoded homologs (*Pedersen and Gerdes, 1999*). Interestingly, most of the *hok*/Sok homologs (*hokA*, *C* and *E*) in *Escherichia coli* are inactivated by the presence of insertion elements located close to the toxin ORF (*Pedersen and Gerdes, 1999*). This observation, together with studies showing a differential expression of several TA systems in response to various stresses (e.g. temperature shift, oxidative stress, starvation) (*Harms et al., 2018*), suggests that chromosomally-encoded TA systems may not be involved in a bactericidal activity, but rather, in a reversible growth arrest in response to a specific stress. Conversely, our results clearly demonstrate that, in line with the bactericidal activity observed for the overexpressed AapA1 toxin (*Arnion et al.,*

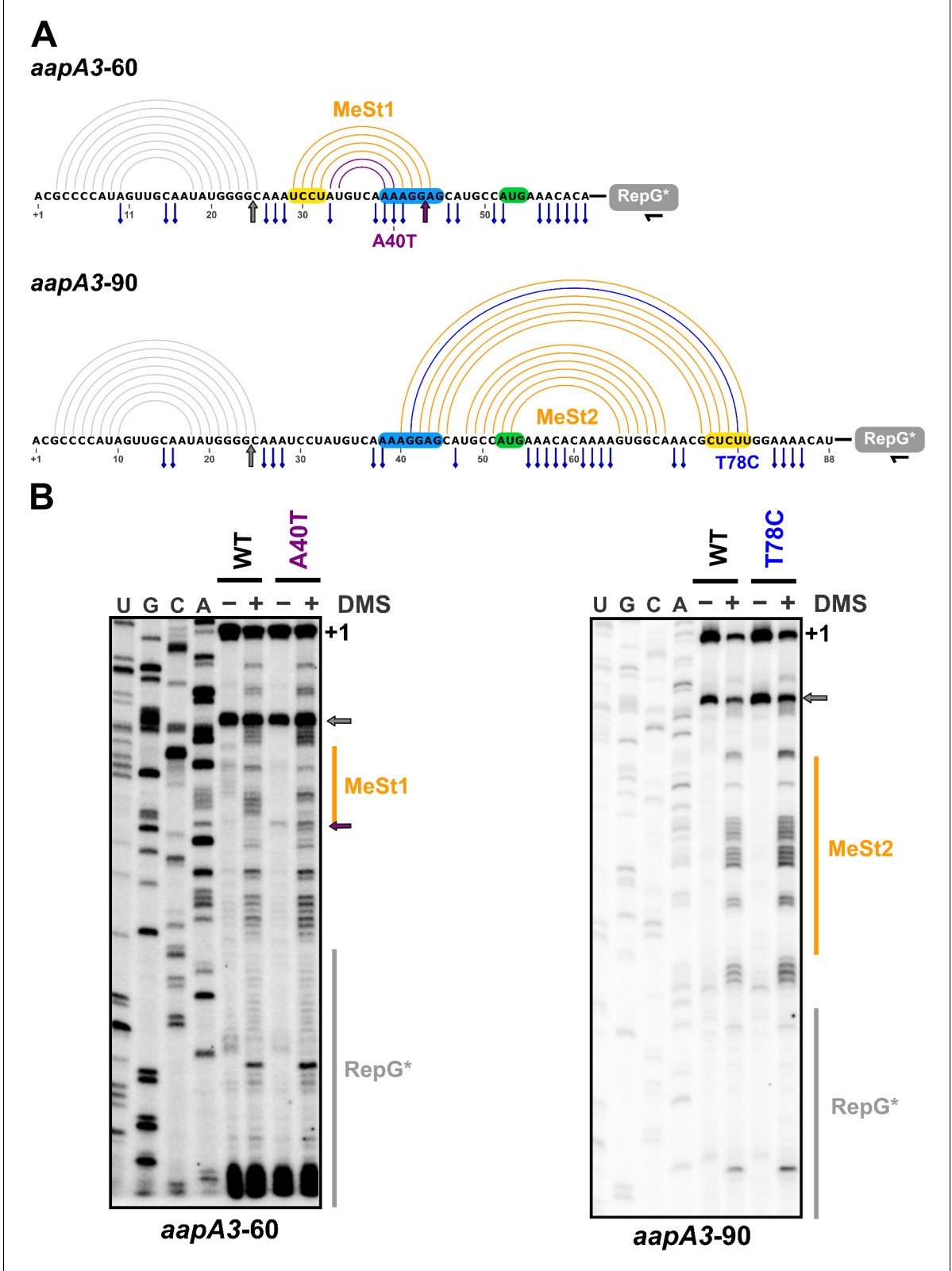

**Figure 8.** In vivo DMS footprinting confirms the existence of the MeSt1 and MeSt2 RNA hairpins. (**A**) Linear representation of the secondary structure mapped in vivo through DMS modification of the *aapA3*-60 and *aapA3*-90 RNA nascent transcripts. The primary sequence of both transcripts is shown, as well as the predicted secondary structure represented by an arc diagram. The position of the point mutations A40T and T78C is shown in purple and blue, respectively. Each RNA intermediate was fused to the RepG non-coding RNA (***Pernitzsch et al., 2014***) to allow a precise transcription

*Figure 8 continued on next page*

*Figure 8 continued*

termination. The RepG sequence (gray box) was mutated (RepG*) to distinguish the chimeric transcript from the endogenous RepG RNA. The horizontal black arrow indicates the position of the oligonucleotide used for reverse extension. The SD and anti-SD sequences are highlighted in blue and yellow, respectively and the start codon is shown in green. The adenine and cytosine bases modified by DMS are indicated by a blue pin. The gray arrow indicates the Reverse Transcriptase (RT) stop observed for both transcripts due to the presence of the 5′ terminal (stable) stem-loop. Interestingly, another RT stop at nt 44 was observed with the *aapA3*-A40T mutant, confirming the stabilizing effect of the A40T mutation (purple arrow). (**B**) DMS-treated (+) and untreated (-) RNA samples were analyzed by primer extension. The reaction products were separated on a 6% PAA denaturing gel. Lanes A, C, G and U correspond to the sequencing ladder of the analyzed region. The arrows indicate the RT stops and the vertical orange lines indicate the position of the metastable hairpins. The RepG* region is shown by a vertical gray line. Note that methylated bases generate RT stops one nucleotide downstream the modified base.

DOI: https://doi.org/10.7554/eLife.47549.026

The following figure supplements are available for figure 8:

**Figure supplement 1.** and *Figure 8—figure supplement 2*: Structural alignment and covariation analysis of *aapA3* mRNA nascent transcript of 45 (1) and 80 nucleotides (2).

DOI: https://doi.org/10.7554/eLife.47549.029

**Figure supplement 2.** Structural alignment and covariation analysis of*aapA3*mRNA nascent transcript of 80 nucleotides.

DOI: https://doi.org/10.7554/eLife.47549.030

**Figure supplement 3.** The two successive *aapA3* mRNA metastable structures have increasing stability and are stabilized by the A40T and the T78C suppressor mutations.

DOI: https://doi.org/10.7554/eLife.47549.028

*2017*), the chromosomal expression of the AapA3 toxin is constitutive and lethal in absence of the IsoA3 antitoxin. Consequently, we took advantage of this lethality to select suppressors and developed the FASTBAC-Seq method to rapidly identify hundreds of intragenic suppressor mutations with nucleotide resolution (*Masachis et al., 2018*).

## A single-nucleotide substitution is sufficient to impede toxin translation

The FASTBAC-Seq method revealed a wide range of unanticipated *cis*-encoded toxicity determinants, affecting either the toxic activity of the protein (described in *Masachis and Darfeuille, 2018*), or its expression (this study). Among the mutations affecting the toxin mRNA expression, we identified five single-nucleotide substitutions able to inhibit the translation of the *aapA3* mRNA without affecting its stability. Three of them were located in the SD sequence. The most highly enriched mutations substituted the guanines at positions 42 and 43 by either an adenine, a cytosine, or a uracil. This revealed 5′-AGG-3′ and 5′–GGA-3′ as the minimal functional SD motifs allowing AapA3 translation, in agreement with the previously identified *H. pylori* SD consensus sequence (5′-AAGGA-3′) (*Sharma et al., 2010*). The third mutation (A40T) was much less enriched, and remarkably, only the transversion from an adenine to a thymine was selected. Another transversion mutation (A28C) was selected 14 nt upstream of the SD sequence. The fact that only transversion mutations were selected at these two positions indicated that the nature of the substituted nucleotide was important, suggesting that they may not directly act at the sequence, but rather at the structure level. Indeed, our results demonstrated that the A28C and A40T mutations create respectively one or two additional base-pair(s) within a stem-loop structure formed by the pairing between the SD sequence and an upstream complementary aSD sequence (aSD1, 5′-UCCU-3′). Destabilizing the A40T-mutated stem by mutating the complementary nucleotide (A33T) restored toxicity, clearly showing that the A40T mutation, despite being located within the SD sequence, acts at the mRNA structural level and not at the sequence level.

Interestingly, the T78C mutation revealed the existence of a second aSD sequence (aSD2) located downstream the SD sequence, within the toxin coding region. This synonymous substitution (UC̲U→UC̲C, Ser codon at position 9) creates a perfect aSD sequence (5′-CUCCU-3′). Although this mutation could potentially create a rare codon reducing toxin translation efficiency, we did not favor this hypothesis since the less frequently used Ser codon in *H. pylori* is UCG (*Lafay et al., 2000*). Interestingly, synonymous mutations close to the translation initiation region (TIR) have been shown to influence gene expression by modulating the stability of mRNA folding rather than by acting at the codon usage level (*Kudla et al., 2009*). In addition, a strong codon bias has also been observed within the first 15 codons, which avoids tight mRNA structure close to the TIR region (*Bentele et al.,*

*2013*; *Bhattacharyya et al., 2018*). Here, we showed that despite the presence of up to thirteen CU-rich sequences in the *aapA3* mRNA, only mutations in the sequences closest to the SD could be selected, reflecting a distance-dependence of these translation regulatory elements. A similar aSD sequence (5'-UCCU-3') has been identified in the coding sequence of the *gnd* gene in *E. coli* (*Carter-Muenchau and Wolf, 1989*). Interestingly, displacing this aSD sequence from its natural position (codon 66) to a more proximal position (codon 13) greatly increased its capacity to inhibit translation.

## Suppressor mutations reveal functional metastable structures acting co-transcriptionally to impede premature toxin translation

The three mutations studied here (A28C, A40T and T78C) act post-transcriptionally after the 3'-end processing by stabilizing SD-sequestering hairpin structures. Importantly, these suppressor mutations do not interfere with the folding pathway of the full-length mRNA, neither affecting its transcription, stability, nor its 3'-end maturation, indicating that they exclusively act on the active, truncated, *aapA3* mRNA form (*Figure 7*). Interestingly, by using the KineFold software, these local hairpins were previously predicted to form during the co-transcriptional folding pathway of several *aapA* mRNAs (*Arnion et al., 2017*; *Xayaphoummine et al., 2005*). Now, our FASTBAC-seq approach reveals that these structures are functional, that is they transiently form during transcription to prevent toxin translation before the full-length transcript is completed. Indeed, stabilizing these hairpins inhibits translation of the *aapA3*-Tr form. This temporal control of gene expression is achieved through the sequential formation of two RNA hairpin structures that mask the SD sequence via CU-rich elements. In the full-length mRNA, these structures are replaced by a more stable one (aSD3) involving an LDI between both ends of the transcript. Similar to the *hok* mRNA, this final mRNA structure is so stable that its translational activation requires a 3'-end processing that removes the aSD3 sequence element. The highly stable structure of the *aapA3*-FL mRNA is also similar to the cloverleaf-like structure present in the 5' UTR of the MS2 coliphage maturation gene (*Groeneveld et al., 1995*). Interestingly, in this case, it may take up to several minutes for the mRNA to be synthesized and properly folded (*van Meerten et al., 2001*), explaining the need of functional transient structural intermediates preventing premature gene expression.

The selection of three stabilizing mutations suggests that the thermodynamic stability of such SD-sequestering stem-loops in the WT context is not sufficient to inhibit the translation of the active *aapA3* mRNA form. Instead, our results suggest that in the WT situation, these SD-sequestering hairpins (MeSt1 and MeSt2, *Figure 7*) are only transiently formed to co-transcriptionally impede premature toxin translation. This transient character is essential to ensure the proper transcription termination and folding of the full-length mRNA, and this is achieved by hierarchically increasing thermodynamic stabilities (*Figure 8—figure supplement 3*). Importantly, the suppressor mutations do not provide enough stabilization to impede the formation of the next most stable structure. Indeed, the A40T-mutated MeSt1 has an energy of −21.10 kcal/mol, while that of the WT MeSt2 is −29.30 kcal/mol (*Figure 8—figure supplement 3*). This may explain why the SD-sequestering mutations do not interfere with the co-transcriptional folding pathway and why the last SD-aSD3 pairing is finally formed in the mutants.

The importance of metastable structures in the *aapA3* mRNA is attested by the strict conservation of the UCU Serine codon at position 9. As our results have shown, a synonymous UCC codon at this position (T78C mutation) would inhibit AapA3 toxin expression, rendering the TA locus non-functional and probably promoting its rapid loss. Our study represents the first in vivo evidence of the existence of sequential RNA metastable structures that avoid, directly but transiently, the co-transcriptional translation of a toxin-encoding mRNA.

The formation of metastable structures has been reported in several RNA-mediated regulatory pathways, including viral RNA replication (*Repsilber et al., 1999*), RNA catalysis (*Pan and Woodson, 1998*), RNA editing (*Mazloomian and Meyer, 2015*), and ribosome biogenesis (*Sharma et al., 2018*). They are usually described as folding intermediates that work in a hierarchical manner to help an RNA molecule reaching its functional conformation (i.e. the most thermodynamically stable conformation). A nice example of such metastable structures has been reported for the regulation of the *hok*/Sok type I TA system in *E. coli*. In this pioneering work, the authors showed that the formation of a metastable hairpin ensures the proper folding of the Hok mRNA into a translationally inert conformation (*Møller-Jensen et al., 2001*; *Nagel et al., 1999*). Although this metastable hairpin is

located at the 5' end of the mRNA, it does not directly mask the SD sequence. Instead, it favors a conformation in which the SD is sequestered by a downstream anti-SD. Other metastable structures are directly involved in the activation or inhibition of gene expression (*Zhu and Meyer, 2015*), as examplified by the structures in the Trp operon leader, the SAM riboswitch and the 5' UTR of the MS2 phage (*Zhu and Meyer, 2015*). The metastable structures of *aapA3* are more reminiscent of the latter example (*van Meerten et al., 2001*), except that in the case of MS2, the transient structure allows translation to occur before the cloverleaf-like structure is formed. Nevertheless, in both cases, a functional transient RNA structure exerts a temporal control of translation, either negatively or positively.

## Conclusion

Although the coupling between transcription and translation in bacteria plays important roles in gene expression (*Kriner et al., 2016*), it can be harmful in the case of toxin-encoding mRNAs. Thus, the metastable RNA structures identified in the present study are essential to uncouple transcription and translation processes and allow the presence of type I TA systems on bacterial chromosomes. Although transient RNA structures can be predicted in silico (*Meyer, 2017a*), their in vivo characterization remains challenging. Several high-resolution methods have been recently devised for analyzing the co-transcriptional folding of regulatory RNAs, both in vitro (*Uhm et al., 2018*; *Watters et al., 2016*) and in vivo (*Incarnato et al., 2017*). These complementary techniques may be useful to analyze the formation of these metastable hairpins in real-time.

# Materials and methods

**Key resources table**

| Reagent type (species) or resources | Designation | Source or reference | Identifiers | Additional information |
|---|---|---|---|---|
| *Chemical compound, drug* | DreamTaq DNA Polymerase | ThermoFischer Scientific | Cat#EP1701 | |
| *Chemical compound, drug* | Phusion High-Fidelity DNA Polymerase | ThermoFischer Scientific | Cat#F530S | |
| *Chemical compound, drug* | PfuUltra High-Fidelity DNA Polymerase | Agilent | Cat#600380 | |
| *Chemical compound, drug* | Alkaline Phosphatase, Calf Intestinal (CIP) | New England Biolabs | Cat#M0290S | |
| *Chemical compound, drug* | T4 Polynucleotide Kinase | New England Biolabs | Cat#M0201S | |
| *Chemical compound, drug* | RNase T1 | New England Biolabs | Cat#AM2283 | |
| *Chemical compound, drug* | *E. coli* RNase H1 | New England Biolabs | Cat#AM2293 | |
| *Chemical compound, drug* | RNasin Ribonuclease Inhibitors | Promega | Cat#N2511 | |
| *Commercial assay or kit* | pGEM-T Easy Vector System | Promega | Cat#A1360 | |
| *Commercial assay or kit* | MEGAScript T7 Kit | ThermoFischer Scientific | Cat#AM1334 | |
| *Commercial assay or kit* | MAXIScript T7 Transciption Kit | ThermoFischer Scientific | Cat#AM1213 | |
| *Commercial assay or kit* | High Purity Plasmid Miniprp Kit | Neo Biotech | Cat#NB-03–0002 | |
| *Commercial assay or kit* | Quick Bacteria Genomic DNA extraction Kit | Neo Biotech | Cat#NB-03–0020 | |
| *Commercial assay or kit* | *E. coli* 30S Extract System for Linear Templates Kit | Promega | Cat#L1030 | |

*Continued on next page*

*Continued*

| Reagent type (species) or resources | Designation | Source or reference | Identifiers | Additional information |
|---|---|---|---|---|
| *Commercial assay or kit* | One Shot TOP10 chemically competent cells | ThermoFischer Scientific | Cat#C404010 | |
| *Recombinant DNA reagent* | Plasmids | This paper | N/A | See *Table 3* for the full list of plasmids used in this study |
| *Sequence-based reagent* | Oligonucleotides | This paper | N/A | See *Table 1* for the full list of oligonucleotides used in this study |
| *Strain, strain background* | E. coli strains | This paper | N/A | See *Table 4* for the full list of *E. coli* strains used in this study |
| *Strain, strain background* | H. pylori strains | This paper | N/A | See *Table 2* for the full list of *H. pylori* strains used in this study |
| *Software, algorithm* | Cutadapt 1.1 | DOI: 10.14806/ej.17.1.200 | https://cutadapt.readthedocs.org/ | |
| *Software, algorithm* | cmpfastq | NIHR Biomedical Research Centre for Mental Health | http://compbio.brc.iop.kcl.ac.uk/software/cmpfastq.php | |
| *Software, algorithm* | Prinseq-lite 0.20.4 | (*Schmieder and Edwards, 2011*) | http://prinseq.sourceforge.net/ | |
| *Software, algorithm* | PANDAseq 2.9 | (*Masella et al., 2012*) | https://github.com/neufeld/pandaseq | |
| *Software, algorithm* | BWA-SW algorithm of BWA 0.7.12 | (*Li and Durbin, 2009*) | https://sourceforge.net/projects/bio-bwa/ | |
| *Software, algorithm* | Samtools 1.2 | (*Li et al., 2009*) | https://sourceforge.net/projects/samtools/ | |
| *Software, algorithm* | Bamtools 2.3.0 | (*Barnett et al., 2011*) | https://github.com/pezmaster31/bamtools | |
| *Software, algorithm* | R 3.2.0 | (R Core Team 2015) | http://www.R-project.org/ | |
| *Software, algorithm* | Differential analyses and Hierarchical tree clustering: Trinity 2.2.0 | (*Haas et al., 2013*) | http://trinityrnaseq.github.io | |
| *Software, algorithm* | Differential analyses: DEseq2 1.10.1 | (*Love et al., 2014*) | http://www.bioconductor.org/packages/release/bioc/html/DESeq2.html | |
| *Software, algorithm* | MAFFT 7.407 | (*Katoh and Toh, 2008*) | https://mafft.cbrc.jp/alignment/software/source.html | |
| *Software, algorithm* | MXSCARNA 2.0 | (*Tabei et al., 2008*) | https://www.ncrna.org/softwares/mxscarna/ | |
| *Software, algorithm* | R-chie | (*Lai et al., 2012*) | https://www.e-rna.org/r-chie/ | |
| *Other* | T1TA database | (Tourasse et al, in preparation) | https://d-lab.arna.cnrs.fr/t1tadb | |
| *Other* | Project and study description | This paper | NCBI BioProject PRJNA497299 | Deposited data |

*Continued on next page*

*Continued*

| Reagent type (species) or resources | Designation | Source or reference | Identifiers | Additional information |
|---|---|---|---|---|
| *Other* | Deep-sequencing datasets raw data | This paper | NCBI SRA SRP166021 | Deposited data |
| *Other* | Single-nucleotide substitutions, number of counts | This paper | NCBI GEO GSE121423 | Deposited data |
| *Other* | Single-nucleotide substitutions, statistical analysis by sequence | This paper | NCBI GEO GSE121423 | Deposited data |
| *Other* | Single-nucleotide substitutions, statistical analysis by position | This paper | NCBI GEO GSE121423 | Deposited data |
| *Other* | Single-nucleotide deletions, statistical analysis by position | This paper | NCBI GEO GSE121423 | Deposited data |
| *Other* | Single-nucleotide insertions, statistical analysis by position | This paper | NCBI GEO GSE121423 | Deposited data |

## Bacterial strains, plasmids and growth conditions

The *H. pylori* strain used in this study is the 26695 reference strain (*Tomb et al., 1997*). Strains were grown on Columbia agar plates supplemented with 7% horse blood and Dent selective supplement (Oxoid, Basingstoke, UK) for 24 to 48 hr depending on the strain. Liquid cultures were performed in Brain-Heart Infusion (BHI) medium (Oxoid) supplemented with 10% fetal bovine serum (FBS) and Dent. *H. pylori* plates and liquid cultures were incubated at 37°C under microaerobic conditions (10% $CO_2$, 6% $O_2$, 84% $N_2$) using an Anoxomat (MART microbiology) atmosphere generator. Plasmids used for cloning were amplified in *E. coli* TOP10 strain, which was grown in Luria-Bertani (LB) media, supplemented either with kanamycin (50 $\mu g.mL^{-1}$), chloramphenicol (30 $\mu g.mL^{-1}$) or ampicillin (100 $\mu g.mL^{-1}$). For *H. pylori* mutant selection and culture, antibiotics were used at the following final concentrations: 20 $\mu g.mL^{-1}$ kanamycine (Sigma), 8 $\mu g.mL^{-1}$ chloramphenicol (Sigma), 10 $\mu g.mL^{-1}$ streptomycin and 10 $\mu g.mL^{-1}$ erythromycin.

## Molecular techniques

Molecular biology experiments were performed according to standard procedures and the supplier's recommendations. High-Purity Plasmid Miniprep Kit (Neo Biotech) and Quick Bacteria Genomic DNA extraction Kit (Neo Biotech) were used for plasmid preparations and *H. pylori* genomic DNA extractions, respectively. PCR were performed either with Dream Taq DNA polymerase (Thermo Fisher Scientific), or with Phusion High-Fidelity Hot Start DNA polymerase (Thermo Fisher Scientific) when the product required high-fidelity polymerase. Site-directed mutagenesis PCR was performed with the PfuUltra High-Fidelity DNA Polymerase (Agilent). All oligonucleotides used in this study are shown in *Table 1*.

## RNA extraction

For RNA extraction, bacterial growth was stopped at the desired $OD_{600nm}$ by adding 650 $\mu l$ cold Stop Solution (95% ethanol, 5% phenol pH 4.5) to 5 ml of culture, which was placed on ice. Cells were then centrifuged for 10 min at 3,500 rpm and 4°C, and the pellets were stored at −80°C. Cell pellets were resuspended in 600 $\mu l$ Lysis Solution (20 mM NaAc pH 5.2, 0.5% SDS, 1 mM EDTA) and added to 600 $\mu l$ hot phenol pH 5.2. After incubation for 10 min at 65°C, the mixture was then centrifuged for 10 min at 13,000 rpm and room temperature. The aqueous phase was next transferred to a phase-locked gel tube (Eppendorf) with an equal volume of chloroform and centrifuged for 10 min at 13,000 rpm and room temperature. Total RNA was then precipitated from the aqueous phase by adding 2.5 volumes of ethanol 100% and 1/10 vol of 3 M NaAc pH 5.2. After centrifugation for 30 min at 13,000 rpm and 4°C, the supernatant was discarded and the pellet was washed with 75% ethanol. Finally, the supernatant was discarded and the RNA pellet air-dried and resuspended in $H_2O$.

For RNA half-life determinations, rifampicin (Sigma, prepared at 34 mg.ml$^{-1}$ in methanol) was added to the culture at a final concentration of 80 μg.ml$^{-1}$ and cells were harvested at the desired time points. A culture where rifampicin was replaced by the same volume of methanol served as a non-treated control.

## Northern blot

For Northern blot analysis, 1 to 10 μg RNA were separated on an 8% polyacrylamide (PAA), 7M urea, 1X Tris Borate EDTA (TBE) gel. RNA was transferred to a nylon membrane (Hybond-N, GE Healthcare Life Science) by electroblotting in TBE 1X at 8V and 4°C overnight. Then, RNA was cross-linked to the membrane by UV irradiation (302 nm) for 2 min in a UV-crosslinker and hybridized with 5′-labeled (γ$^{32}$P) oligodeoxynucleotides in a modified Church Buffer (1 mM EDTA, 0.5 M NaPO$_4$ pH 7.2, 7% SDS) overnight at 42°C. Membranes were washed two times 5 min in 2X SSC, 0.1% SDS, and revealed using a Pharos FX phosphorimager (Biorad). For riboprobes, a DNA template containing a T7 promoter sequence was amplified by PCR from *H. pylori* 26695 genomic DNA as template. In vitro transcription was performed as described in the MaxiScript T7 Transcription Kit (Ambion) in the presence of 50 μCi of $^{32}$P-α-UTP and 1 mM cold UTP and purified on a Sephadex G25 column (GE Healthcare). Hybridization was performed in the modified Church Buffer at 65°C and the membrane was washed two times 5 min in 2X SSC, 0.1% SDS at 65°C. For the detection of *aapA3* mRNA species the $^{32}$P-labeled primer FD38 was used. To detect the *aapA3* mutants sequestering the SD region (where the primer FD38 binds), a riboprobe corresponding to the 5′ UTR of the mRNA was transcribed from a PCR fragment containing the T7 promoter and amplified with the FA170/FA11 primer pair. IsoA3 RNA was detected with a riboprobe corresponding to the *aapA3*-Tr RNA species transcribed from a PCR fragment containing the T7 promoter and amplified with the primer pair FA170/FA173.

## In vitro transcription and translation assays

For in vitro synthesis of the aapA3 and IsoA3 RNAs, DNA templates were amplified from *H. pylori* 26695 genomic DNA using primer pairs FA170/FA175 (aapA3-FL), FA170/FA173 (aapA3-Tr), and FD11/FD17 (IsoA3), each forward primer carrying a T7 promoter sequence (see primer list, *Table 1*). In vitro transcription was carried out using the MEGAscript T7 Transcription Kit (Ambion #AM1334) according to the manufacturer's protocol. After phenol:chloroform extraction followed by isopropanol precipitation, the RNA samples were desalted by gel filtration using a Sephadex G-25 (GE Healthcare) column. For in vitro translation of the aapA3-FL and aapA3-Tr mRNAs, 0.5 μg of RNA was added to the *E. coli* S30 Extract System for Linear Templates Kit (Promega #L1030) as previously described (*Sharma et al., 2010*).

## In vitro structure probing

20 pmol of both *aapA3-FL* and *aapA3-Tr* transcripts were dephosphorylated with 10 U of calf alkaline phosphatase (CIP) at 37°C for 1 hr. RNA was isolated by phenol extraction and precipitated overnight at −20°C in the presence of 30:1 ethanol: 0.3M NaOAc pH 5.2 and 20 μg GlycoBlue. The dephosphorylated RNA was then 5′ end-labeled with 10 pmol $^{32}$P-γ-ATP using the T4 polynucleotide kinase (PNK) for 30 min at 37°C. Unincorporated nucleotides were removed using a MicroSpin G-25 column and labeled RNA was purified on an 8% PAA gel containing 7 M urea and 1X TBE. Upon visualization of the labeled RNA, the band corresponding to the RNA species of interest was cut from the gel and eluted overnight at 4°C under shaking in 750 μl RNA elution buffer (0.3M NH$_4$Ac, 0.1% SDS, 1 mM EDTA). RNA was extracted by Phenol/Chloroform/Isoamyl alcohol (25:24:1 v/v), and precipitated by ethanol (2.5V), pellets were washed and resuspended in 50 μl H$_2$O and stored at −20°C.

Before use, each in vitro transcribed RNA was denatured by incubation at 90°C for 2 min in the absence of magnesium and salt, then chilled on ice for 1 min, followed by a renaturation step at room temperature for 15 min in 1X Structure Buffer (10 mM Tris-HCl pH 7.0, 10 mM MgCl$_2$, 100 mM KCl). Structure probing analyses were performed as described previously (*Darfeuille et al., 2007*; *Sharma et al., 2007*; *Sharma et al., 2010*), using 0.1 pmol of 5′ end-labeled RNA. To determine the secondary structure of RNA, 1 μl RNase T1 (0.01 U.μl-1; Ambion) was added to the labeled RNA and incubated in 1X Sequencing Buffer (20 mM Sodium Citrate, pH 5.0, 1 mM EDTA, 7M Urea) for 5 min

**Table 1.** Oligonucleotides used in this work.

| Name | Sequence (5'→3' direction) | Used for |
|------|---------------------------|----------|
| FD11 | GAAATTAATACGACTCACTAT AGCAAGAGCGTTTGCCACTT | Reverse primers carrying a T7 promoter for IsoA3 amplification for in vitro transcription |
| FD17 | ACGCCCCATAGTTGCAATAT | Forward primer for IsoA3 amplification for in vitro transcription |
| FD35 | TCGGAATGGTTAACTGGGTAGTTCCT | Reverse primer for 5S rRNA mRNA detection by Northern Blot |
| FD38 | GCTCCTTTTGACATAGGATT | Reverse primer for *aapA3* mRNA detection by Northern Blot |
| FA110 | TGCTTTATAACTATGGATTAAAC | Forward primer for *rpsL-erm* cassette amplification from pSP60 |
| FA111 | TTACTTATTAAATAATTTATAGC | Revese primer for *rpsL-erm* cassette amplification from pSP60 |
| FA170 | GAAATTAATACGACTCACTATAG GACGCCCCATAGTTGCAATAT | Forward primer carrying a T7 promoter for *aapA3* in vitro transcription |
| FA173 | AGGAAACCCCTAAGCTTAAAAGC | Reverse primer for *aapA3*-Tr amplification |
| FA175 | GACCAACGCCCCAAAAGTC | Reverse primer for *aapA3* full-length amplification |
| FA281 | AGCATGCCATTAAACACAAA | Forward primer for mutagenesis of *aapA3* 26695 start codon (G54A) |
| FA282 | TTTGTGTTTAATGGCATGCT | Reverse primer for mutagenesis of aapA3 26695 start codon (G54A) |
| FA283 | TGGAAAACCTTGTACTTTGAGT | Forward primer for mutagenesis of IsoA3 −10 box: mutations A87C/A90G |
| FA284 | ACTCAAAGTACAAGGTTTTCCA | Reverse primer for mutagenesis of IsoA3 −10 box: mutations A87C/A90G |
| FA386 | CCAAGAGCGTTTGCCACTTTTG | Reverse primer for *aapA3*/IsoA3 locus split cloning in pGEMT (upstream fragment) |
| FA387 | CACAAAAGTGGCAAACGCTC | Forward primer for *aapA3*/IsoA3 locus split cloning in pGEMT (downstream fragment) |
| FA395 | <u>CTTTCCCTACACGACGCTCTTCCGATCT</u>CTATCCAATAAAGATAAGC | Forward primer for *aapA3* 26695 amplification for Illumina paired-end sequencing |
| FA396 | <u>GGAGTTCAGACGTGTGCTCTTCCGATC T</u>GCACTCTATGAGGGGATTTAG | Reverse primer for *aapA3* 26695 amplification for Illumina paired-end sequencing |
| FA406 | GCATTATAAAATGAAATCC | Forward primer for the amplification of *aapA3* 26695 fragment Up from *hpn*-like |
| FA407 | **GTTTAATCCATAGTTATAAAGCA**CAAAAAGAGGGATTTTAAAAG | Reverse primer for the amplification of *aapA3* 26695 Up fragment to generate the *aapA3*/IsoA3 locus deletion designed for deep-seq |
| FA408 | **GCTATAAATTATTTAATAAGTAA**CCGCTTGCTCTAGCTTTTTG | Forward for the amplification of *aapA3* 26695 Down fragment to generate the *aapA3*/IsoA3 locus deletion designed for deep-seq |
| FA409 | CTAGCCACGCTCTATTAGAG | Reverse for the amplification of *aapA3* 26695 Down fragment to generate the *aapA3*/IsoA3 locus deletion designed for deep-seq |
| FA465 | CAATATGGGGCAAcTCCTATGTC | Forward primer for the introduction of the suppressor A28C |

*Table 1 continued on next page*

*Table 1 continued*

| Name | Sequence (5'→3' direction) | Used for |
|---|---|---|
| FA466 | GACATAGGAgTTGCCCCATATTG | Reverse primer for the introduction of the suppressor A28C |
| FA467 | CCTATGTCAAtAGGAGCATG | Forward primer for the introduction of the suppressor A40T |
| FA468 | CATGCTCCTaTTGACATAGG | Reverse primer for the introduction of the suppressor A40T |
| FA511 | CAAAAGTGGCAAACGCTCc TGGAAAACcTTgTACTTTGAGTTTG | Forward primer for the introduction of the suppressor T78C |
| FA512 | GTTTTCCAgGAGCGTTT GCCACTTTTG | Reverse primer for the introduction of the suppressor T78C |
| FA535 | CCTATGTCAAAAGaAGC ATGCCATGAAACAC | Forward primer for the introduction of the SD mutation G43A |
| FA536 | GTGTTTCATGGCATGCT tCTTTTGACATAGG | Reverse primer for the introduction of the SD mutation G43A |
| FA546 | GTTGCAATATGGGGCAAATC CTtTGTCAAAAGGAGCATGCC | Forward primer for the introduction of A33T mutation (complementation of A40T suppressor) |
| FA547 | GGCATGCTCCTTTTGACAaA GGATTTGCCCCATATTGCAAC | Reverse primer for the introduction of A33T mutation (complementation of A40T suppressor) |
| FA548 | GTTGCAATATGGGGCAAAT CCTtTGTCAAtAGGAGCATGCC | Forward primer for the introduction of A33T mutation in *aapA3* A40T mutant background |
| FA549 | GGCATGCTCCTaTTGACAa AGGATTTGCCCCATATTGCAAC | Reverse primer for the introduction of A33T mutation in *aapA3* A40T mutant background |
| FA633 | CATGGCATGCTCCTTT | RNaseH/oligonucleotide accessibility assay on WT-FL, WT-Tr and T78C-Tr *aapA3* mRNAs |
| FA644 | CATAGGATTTGCCCCA | RNaseH/oligonucleotide accessibility assay on A40T-Tr aapA3 mRNA |
| FA651 | CAAAGGATTTGCCCCA | RNaseH/oligonucleotide accessibility assay on A33T/A40T-Tr *aapA3* mRNA |
| FA652 | CATAGGAGTTGCCCCA | RNaseH/oligonucleotide accessibility assay on A28C-Tr *aapA3* mRNA |
| FA786 | *CCATAAGGAATGGTTGGAC*GTG TTTCATGGCATGCTCCTTTTG | Reverse primer to fuse repG[*] (sequence in italics) downstream nt 1–60 of *aapA3*. |
| FA787 | *CCATAAGGAATGGTTGGAC*GTGT TTCATGGCATGCTCCTaTTG | Reverse primer to fuse repG[*] (sequence in italics) downstream nt 1–60 of the A40T mutated *aapA3* |
| FA789 | *CCATAAGGAATGGTTGGAC*TA ATGTTTTCCAAGAGCGTTTG | Reverse primer to fuse repG[*] (sequence in italics) downstream nt 1–90 of *aapA3*. |
| FA790 | *CCATAAGGAATGGTTGGAC*TAA TGTTTTCCAgGAGCGTTTG | Reverse primer to fuse repG[*] (sequence in italics) downstream nt 1–90 of the T78C mutated *aapA3* |
| FA791 | *CTTGGCGGTTGTTGTTTTTTC* CGCTTGCTCTAGCTTTTTG | Forward primer containing the 3' end of RepG[*] (sequence in italics) to amplify *aapA3* Down fragment |
| FA794 | GTCCAACCATTCCTTATGG | Forward primer to amplify RepGstar[#] |
| FA795 | AAAAAACAACAACCGCCAAG | Reverse primer to amplify RepGstar[#] |

[*]Nucleotide positions are indicated relative to the AapA3 transcriptional start site (TSS, +1).

[**]Sequences highlighted in bold correspond to *rpsL-erm* 5'-overhang tails used for assembly PCR during the construction of the *aapA3/IsoA3* deleted strain.

[***]Underlined sequences correspond to the DNA adaptors used for Illumina paired-end sequencing approach.

[****]Nucleotides in lowercase correspond to mutations introduced by site-directed mutagenesis PCR.

#the sequence of RepGstar used in this study was (underlined nucleotides are indicating the mutations introduced to distinguish RepG from the endogenous copy; **Pernitzsch et al., 2014**): GTCCAACCATTCCTTATGGTTTGGTTGGAACCGCTTAAGATTGAAGGGTCA<u>ACT</u>A<u>CC</u>A<u>CT</u>CCTTTCCCTTTGTCTTGGCGGTTGTTGTTTTTTGGATCC
DOI: https://doi.org/10.7554/eLife.47549.020

at 37°C. Lead acetate (5 mM final concentration) digestions of both *aapA3*-Tr and *aapA3*-FL were done in the absence or in the presence of 2–10-fold excess of cold IsoA3 RNA. All reactions were stopped by adding 10 µl of 2X Loading Buffer (95% formamide, 18 mM EDTA, Xylene Blue and Bromophenol Blue. Cleaved fragments were then analyzed on an 8% denaturing PAA gel containing 7M urea and 1X TBE. Gels were dried for 45 min at 80°C, and revealed using a Pharos FX phosphorimager (Biorad).

## RNase H1/oligonucleotide accessibility assay

Internally-labeled transcripts were in vitro-transcribed using the MAXIscript T7 Transcription Kit (Ambion #AM1312) in presence of 2.2 µM α-$^{32}$P-UTP according to the manufacturer's protocol. Labeled RNA was purified on an 8% PAA gel containing 7 M urea and 1X TBE, eluted overnight at 4°C under shaking in 750 µl elution buffer (0.1 M NaOAc pH 5.2, 0.1% SDS). RNA was desalted and concentrated by ethanol precipitation, pellets were resuspended in 100 µl H$_2$O. Approximately 30 fmol of RNA were used for RNase H/oligonucleotide accessibility assays. Before use, each in vitro-transcribed RNA and DNA oligonucleotides were denatured as described for structure probing. Next, DNA oligonucleotides complementary to the region around the SD sequence (FA633 for WT and T78C (aSD2) mRNA; FA644 for A40T mRNA (aSD1); FA651 for the double mutant A33T/A40T mRNA (aSD1); and FA652 for A28C mRNA) were added to a final concentration of 0 to 10 µM. Reactions were adjusted to a final volume of 10 µl with H$_2$O and incubated for 30 min at 30°C in the presence or absence (control) of 0.25 U *E. coli* RNase H1 (Ambion #AM2293). Reactions were then stopped by addition of 10 µl of 2X Loading Buffer (95% formamide, 18 mM EDTA, Xylene Blue and Bromophenol Blue). Cleaved fragments were analyzed on an 8% denaturing PAA gel containing 7M urea and 1X TBE. Gels were dried 45 min at 80°C, and revealed using a Pharos FX phosphorimager (Biorad).

## *H. pylori* chromosomal manipulation techniques

All mutant *H. pylori* strains listed in *Table 2* were generated by chromosomal homologous recombination of PCR-generated constructs, introduced by natural transformation, as previously described (*Masachis et al., 2018*). In all cases, constructs contained ≈ 400 nt of the up- and downstream chromosome regions of the target gene, flanking the DNA fragment to be introduced (i.e. antibiotic resistance marker to generate deletions or a WT (for the complementation strain, C$^{A3}$) or mutated copy of the target gene). DNA fragments of interest were previously cloned in *E. coli* vectors to avoid *H. pylori* WT genomic DNA (gDNA) contamination (see '*aapA3*/IsoA3 locus sub-cloning in *E. coli*' section below). Constructs were generated by PCR assembly of PCR products amplified from the plasmids listed in *Table 3* with the oligonucleotides shown in *Table 1*. For strains H321, H322, H324 and H325, PCR assembly was performed to introduce the repG* sequence downstream of nt 60 or nt 90 of *aapA3*. WT or single point mutants (A40T and T78C) variants of *aapA3* were used. Prior to transformation, *H. pylori* strains (number of cells corresponding to 1 OD$_{600nm}$) were grown on non-selective CAB plates. After 4 hr incubation at 37°C under microaerobic conditions, 1 µg of PCR assembly product was added to the cells and plates were incubated for another 16 hr. Transformed cells were then selected on plates supplemented with the appropriate antibiotics and incubated for 4–6 days until isolated colonies appeared. Genomic DNA from transformants was purified using the Quick Bacteria Genomic DNA extraction Kit and subjected to PCR and Sanger sequencing for mutant validation.

## Deletion of the *aapA3*/IsoA3 locus using the *rpsL$_{Cj}$-erm* counterselection marker

The counterselection cassette *rpsL$_{Cj}$-erm* was used to generate an *H. pylori* 26695 strain deleted for the *aapA3*/IsoA3 locus following the protocol described in *Masachis and Darfeuille (2018)*. First, the 26695 *H.. pylori* strain used in this study was modified in order to become resistant to

**Table 2.** *Helicobacter pylori* strains used in this work.

| Name | Strain number | Description | Plasmid | Resistance | Reference |
|---|---|---|---|---|---|
| 26695 | JR34 (H5) | Wild type 26695 strain, Institut Pasteur collection, CIP106780 | none | - | (*Tomb et al., 1997*) |
| 26695 *rpsL*K43R | H158 | *rpsL* gene mutated on the Lys at position 43 to Arg (K43R) | none | Str$^R$ | This study |
| 26695 Δ*aapA3*/IsoA3 | H204 | Δ*aapA3*/IsoA3::*rpsL$_{Cj}$*-erm/*rpsL*K43R | none | Erm$^R$ | This study |
| 26695 Complemented *aapA3*/IsoA3 | H170 | Δ*aapA3*/IsoA3 + *aapA3*/IsoA3 (C$^{A3}$) | none | Str$^R$ | This study |
| 26695 *aapA3 start* | H171 | *aapA3* start codon mutated to ATT by the single point mutation G54T | none | Str$^R$ | This study |
| 26695 *aapA3* ΔT109 | H172 | *aapA3* carrying a −1 frameshift mutation (deletion of T at position 109) generating a 23 amino acids-long peptide | none | Str$^R$ | This study |
| 26695 *aapA3 start*/pIsoA3* | H173 | *aapA3* G54T and IsoA3 promoter inactivated by the double point mutation A87C/A90G | none | Str$^R$ | This study |
| 26695 *aapA3* G43A/pIsoA3* | H247 | *aapA3* SD inactivated by the G43A mutation and IsoA3 promoter A87C/A90G | none | Str$^R$ | This study |
| 26695 *aapA3* T107C/pIsoA3* | H278 | *aapA3* ORF suppressor T107C (Phe 19 Ser) and IsoA3 promoter A87C/A90G | none | Str$^R$ | This study |
| 26695 *aapA3* A28C/pIsoA3* | H224 | *aapA3* A28C suppressor mutation and IsoA3 promoter A87C/A90G | none | Str$^R$ | This study |
| 26695 *aapA3* A40T/pIsoA3* | H225 | *aapA3* A40T and IsoA3 promoter A87C/A90G | none | Str$^R$ | This study |
| 26695 *aapA3* A33T/A40 T/G54T/ pIsoA3* | H257 | *aapA3* A40T/A33T/G54T and IsoA3 promoter A87C/A90G | none | Str$^R$ | This study |
| 26695 *aapA3* T78C/pIsoA3* | H240 | *aapA3* T78C and IsoA3 promoter A87C/A90G | none | Str$^R$ | This study |
| 26695 *aapA3* T78C | H226 | *aapA3* T78C with wild-type IsoA3 expression | none | Str$^R$ | This study |
| 26695 *aapA3-60_WT* | H321 | Nt 1–60 of *aapA3* fused to repG* | none | Str$^R$ | This study |
| 26695 *aapA3-60_A40T* | H322 | Nt 1–60 of *aapA3* carrying the A40T mutation fused to repG* | none | Str$^R$ | This study |
| 26695 *aapA3-90_WT* | H324 | Nt 1–90 of *aapA3* fused to repG* | none | Str$^R$ | This study |
| 26695 *aapA3-90_T78C* | H325 | Nt 1–90 of *aapA3* carrying the T78C mutation fused to repG* | none | Str$^R$ | This study |

*Nucleotide positions are indicated relative to the AapA3 transcriptional start site (TSS, +1).

DOI: https://doi.org/10.7554/eLife.47549.031

streptomycin. To this end, we introduced by homologous recombination a mutation (K43R) in the *rpsL* gene coding for the small S12 ribosomal protein (*Masachis et al., 2018*). Then, up- and downstream fragments of the locus were amplified with the primer pairs FA406/FA407 and FA408/FA409. These flanking regions (415 and 418 nt-long, respectively) allow chromosomal homologous recombination to occur. The internal primers (FA407 and FA408) were used to introduce a 3'- and 5'- *rpsL$_{Cj}$*-erm cassette homology tail, respectively, to allow subsequent PCR assembly. The *rpsL$_{Cj}$-erm* cassette was amplified from the pSP60-2 plasmid (*Table 3*) using the primer pair FA110/FA111. Then, the up- and downstream fragments were assembled with the *rpsL$_{Cj}$-erm* cassette by PCR assembly

**Table 3.** Plasmids used in this work.

| Name | Description | Origin/ Marker | Reference |
|---|---|---|---|
| pSP60 −2 | pSP60 carrying the counter selection cassette *rpsL-erm* | pSC101[*]/ Amp[R] | (*Dailidiene et al., 2006*; *Pernitzsch et al., 2014*) |
| pA3-Up WT | pGEM-T carrying the upstream fragment of the *aapA3*/IsoA3 locus amplified with the FA406/FA386 primer pair | ColE1/ Amp[R] | This study |
| pA3-Down WT | pGEM-T carrying the downstream fragment of the *aapA3*/IsoA3 locus amplified with the FA409/FA387 primer pair | ColE1/ Amp[R] | This study |
| pA3-Down pIsoA3[*] | pGEM-T carrying the downstream fragment of the *aapA3*/IsoA3 locus containing IsoA3 −10 box inactivated (A87C/A90G) | ColE1/ Amp[R] | This study |
| pA3-Up *start* | pGEM-T carrying the upstream fragment of the *aapA3*/IsoA3 locus containing the AapA3 start codon mutation G54T | ColE1/ Amp[R] | This study |
| pA3-Up A28C | pGEM-T carrying the upstream fragment of the *aapA3*/IsoA3 locus containing the suppressor A28C | ColE1/ Amp[R] | This study |
| pA3-Up A33T | pGEM-T carrying the upstream fragment of the *aapA3*/IsoA3 locus containing the A33T mutation | ColE1/ Amp[R] | This study |
| pA3-Up A40T | pGEM-T carrying the upstream fragment of the *aapA3*/IsoA3 locus containing the suppressor A40T | ColE1/ Amp[R] | This study |
| pA3-Up A40T/A33T | pGEM-T carrying the upstream fragment of the *aapA3*/IsoA3 locus containing the A40T and the compensatory mutation A33T | ColE1/ Amp[R] | This study |
| pA3-Up T78C | pGEM-T carrying the upstream fragment of the *aapA3*/IsoA3 locus containing the suppressor T78C | ColE1/ Amp[R] | This study |
| pA3-Down T78C | pGEM-T carrying the downstream fragment of the *aapA3*/IsoA3 locus containing the suppressor T78C | ColE1/ Amp[R] | This study |
| pA3-Down T78C/pIsoA3[*] | pGEM-T carrying the downstream fragment of the *aapA3*/IsoA3 locus containing the suppressor T78C and IsoA3 −10 box inactivated (A87C/A90G) | ColE1/ Amp[R] | This study |
| pA3-Up G43A | pGEM-T carrying the upstream fragment of the *aapA3*/IsoA3 locus containing the SD suppressor G43A | ColE1/ Amp[R] | This study |

[*]Nucleotide positions are indicated relative to the AapA3 transcriptional start site (TSS, +1).
DOI: https://doi.org/10.7554/eLife.47549.032

using the external primers (FA406/FA409) (see *Figure 1—figure supplement 2* and primer list in *Table 1*). This construct (1294 nt-long) was used to perform natural *H. pylori* transformation by homologous recombination, as previously described (*Bury-Moné et al., 2001*). This process generated the strain that will serve as recipient in all our successive transformation experiments, Δ*aapA3*/IsoA3::*rpsL*$_{Cj}$-*erm*/K43R (*Table 2*).

## *aapA3*/IsoA3 locus sub-cloning in *E. coli*

Because *H. pylori* has a highly active homologous recombination machinery, a cloning step of the *aapA3*/IsoA3 locus in an *E. coli* vector was essential to avoid contamination with WT *H. pylori* gDNA of the PCR products used in the transformation assays. To this end, the *aapA3*/IsoA3 locus was split into two fragments amplified with the Phusion High-Fidelity Hot Start DNA Polymerase and the primer pairs FA406/FA386 ('Up' fragment of 638 nt containing 415 nt of homology region, the *aapA3* promoter and the first 10 codons of the AapA3 ORF, *Figure 1—figure supplement 2*) and FA409/FA387 ('Down' fragment of 680 nt containing IsoA3 promoter, the rest of *aapA3* mRNA and

**Table 4.** *Escherichia coli* strains used in this work.

| Name | Description/genotype | Plasmid | Resistance | Reference |
|------|---------------------|---------|-----------|-----------|
| TOP10 | *mcrA Δ(mrr-hsdRMS-mcrBC)* Φ80*lacZ*ΔM15 Δ*lacX74 deoR recA1 araD139 Δ(ara-leu)* 7697 *galU galK rpsL endA1 nupG* | none | none | Thermo Fisher Scientific, Invitrogen |
| A3-Up WT | TOP10 | pA3-Up WT | Amp$^R$ | This study |
| A3-Down WT | TOP10 | pA3-Do WT | Amp$^R$ | This study |
| A3-Down pIsoA3* | TOP10 | pA3-Do pIsoA3* | Amp$^R$ | This study |
| A3-Up *start* | TOP10 | pA3-Up *start* | Amp$^R$ | This study |
| A3-Up A28C | TOP10 | pA3-Up A28C | Amp$^R$ | This study |
| A3-Up A33T | TOP10 | pA3-Up A33T | Amp$^R$ | This study |
| A3-Up A40T | TOP10 | pA3-Up A40T | Amp$^R$ | This study |
| A3-Up A40T/A33T | TOP10 | pA3-Up A40T/A33T | Amp$^R$ | This study |
| A3-Up T78C | TOP10 | pA3-Up T78C | Amp$^R$ | This study |
| A3-Down T78C | TOP10 | pA3-Do T78C | Amp$^R$ | This study |
| A3-Down T78C/pIsoA3* | TOP10 | pA3-Do T78C/pIsoA3* | Amp$^R$ | This study |
| A3-Up G43A | TOP10 | pA3-Up G43A | Amp$^R$ | This study |

*Nucleotide positions are indicated relative to the AapA3 transcriptional start site (TSS, +1).

DOI: https://doi.org/10.7554/eLife.47549.033

418 nt of homology region, *Figure 1—figure supplement 2*). Note that the FA386 and FA387 primers have 25 nucleotides of overlap to allow PCR assembly. Each fragment was cloned in a separate pGEM-T (Promega) plasmid (*Table 3*) and transformed into One-Shot TOP10 chemically competent *E. coli* cells (see Experimental Models in the KEY RESOURCES TABLE and *Table 4*.

## Mutant generation by site-directed mutagenesis PCR

Plasmids and custom-designed overlapping oligonucleotides containing the desired mutations were used for site-directed mutagenesis PCR using the PfuUltra high-fidelity DNA polymerase. To inactivate the IsoA3 −10 box, two synonymous point mutations (adenines +87 and +90 from the toxin transcription start site [TSS] were mutated to guanine and cytosine, respectively) were introduced using the primer pair FA283/FA284 see *Figure 1* for details). This strategy allowed us to preserve the toxin coding sequence while completely abolishing the transcription of the antitoxin, as previously shown (*Arnion et al., 2017*). To inactivate the toxin start codon, a single point mutation in the third codon position was introduced (guanine 54 was mutated to thymine) using the primer pair FA281/FA282 (see *Figure 1* for details). WT or mutated fragments were amplified from the previously generated plasmids using the same primer pairs as those used for insert amplification prior to cloning. PCR assembly with 35 amplification cycles, the Phusion High-Fidelity Hot Start DNA Polymerase and the external primers FA406/FA409 was performed to generate the *aapA3*/IsoA3 locus variants (1294-nt amplicon) that were subsequently used as DNA substrates for *H. pylori* natural transformation. For the in vivo validation of the suppressor mutants studied here, the same protocol was used adapting the DNA oligonucleotides containing the desired mutations.

## Determining *H. pylori* transformation efficiency

For the transformation assays aiming at the determination of the transformation efficiency as an indirect proof of the toxicity of the expression of a PCR construct, transformation patches (after 16 hr growth upon DNA addition) were recovered and resuspended in 1 mL BHI. Ten-fold serial dilutions adapted to each transformation case ($10^7$, $10^6$ and $10^5$ for non-selective media; and $10^4$, $10^3$, $10^2$ for selective media upon transformation with water or a toxic construct; and $10^5$, $10^4$, $10^3$ for selective media upon transformation with non-toxic constructs) were performed. Allelic replacement events were selected by the use of streptomycin-containing plates (selection of loss of the *rpsL$_{Cj}$-erm* cassette, Str$^R$). The number of Str$^R$CFU/total CFU was calculated, plotted and statistically analyzed by unpaired *t* (student)-test (GraphPad Prism software version 7).

## *H. pylori* transformation assay to identify toxicity suppressors by illumina sequencing

Transformation assays to select toxicity suppressors were performed in three biological replicates using the wild-type (WT) or antitoxin promoter inactivated PCR-generated constructs (pIsoA3*). Upon transformation, all bacteria were recovered and serially diluted. Transformants were selected on streptomycin-containing plates by using optimized dilutions (nine plates/replicate of $10^1$ dilution for pIsoA3* and three plates/replicate of $10^3$ dilution for WT). Three days after transformation, colonies were pooled (approximately 60,000 colonies per transformation) and genomic DNA was extracted. Next, the *aapA3*/IsoA3 locus was amplified with the primer pair FA395/FA396 (426-nt amplicon, *Figure 1—figure supplement 2*), which allows the introduction of the DNA adapters for Illumina paired-end sequencing. Importantly, to avoid amplification from phenotypic revertant clones (mutated in the *rpsL* gene), the FA395 and FA396 primers are nested to the ones used for locus deletion (FA407 and FA408, *Figure 1—figure supplement 2*), thus, binding to deleted regions that are re-introduced only upon recombination. For this PCR, the Phusion High-Fidelity Hot Start DNA polymerase (Thermo Fisher) and 35 amplification cycles were used. Finally, the samples were sent for sequencing at the Platforme GeT-PlaGe-, Genotoul Centre INRA, Toulouse, France. Sequencing was done on an Illumina MiSeq instrument in paired-end mode 2 × 250 nt (overlapping reads).

## Polysome fractionation in sucrose gradients

*H. pylori* strains were grown as described above. At an early exponential phase (OD$_{600nm}$ <0.9), chloramphenicol (100 μg.mL-1) was added to the culture to stabilize translating ribosomes. After 5 min incubation at 37˚C, cultures were quickly cooled by transferring them into pre-chilled flasks immerged in a dry ice/ethanol bath. Cultures were then centrifuged for 10 min at 3,500 rpm and 4˚C and pellets were washed with Buffer A (10 mM Tris-HCl pH 7.5; 60 mM KCl; 10 mM MgCl$_2$) and frozen at −80˚C. Then, pellets were resuspended in 500 μl of Buffer A containing RNasin Ribonuclease Inhibitor (Promega) and cells were lyzed with glass beads in a Precellys homogenizer (Bertin). Lysates were recovered and immediately frozen in liquid nitrogen. About 10 OD$_{260}$ units of lysate were layered onto 10–40% sucrose gradients in Grad-Buffer (10 mM Tris-HCl pH 7.5; 50 mM NH$_4$Cl; 10 mM MgCl$_2$; 1 mM DTT) and centrifuged at 35,000 rpm for 3.75 hr at 4˚C in a SW41 Ti rotor. Gradients were analysed with an ISCO UA-6 detector with continuous OD monitoring at 254 nm. Fractions of 500 μl were collected and RNA was precipitated overnight at −20˚C in the presence of 1 vol of ethanol containing 150 mM of sodium acetate (pH 5.2). RNA was extracted and subjected to Northern Blot analysis following the protocols described above.

## In vivo DMS footprinting

*H. pylori* strains H321 (WT60), H322 (60_A40T), H324 (WT90), and H325 (90_T78C) were grown in BHI + 10% FBS medium at 37˚C until OD$_{600nm}$ reached 0.5. Aliquots of 25 ml were then pelleted by centrifugation at 3000 rpm for 15 min at room temperature. In vivo DMS modification was carried out as described in *Incarnato et al. (2017)* with minor modifications. Cell pellets were resuspended in 1 ml of structure probing buffer (50 mM HEPES-KOH pH 7.9; 100 mM NaCl; 3 mM KCl). DMS (Sigma Aldrich, cat. D186309) was diluted 1:3 in 100% ethanol to a final concentration of 3.52 M. Diluted DMS was added to bacteria to a final concentration of 200 mM. Samples were incubated with moderate shaking (800 rpm) at 25˚C for 2 min, then immediately transferred to ice. DTT was added to a final concentration of 0.7 M to quench DMS, and samples were vigorously vortexed for 10 s. Bacteria were then pelleted by centrifugation at 13 000 rpm and 4˚C for 30 s and supernatants were discarded. Pellets were then washed once with 1 ml Isoamyl alcohol (Merck, cat. 1009791000) to remove traces of DMS. After another centrifugation, pellets were snap-frozen in liquid nitrogen and stored at − 80˚C. Total RNA was prepared as described above. Primer extensions were carried out as in *Iost et al. (2019)* with minor modifications. After denaturation at 90˚C, 5 μg of total RNA and 0.2 pmoles of 5' end-labeled primer (FD861) were annealed. Reverse transcriptions were performed with Superscript II (Invitrogen, cat. 18064–022) at 42˚C for 30 min. The extension products were then separated on 6% polyacrylamide/7 M urea gels and detected by autoradiography using a Pharos FX phosphorimager (Biorad). Sequencing reactions were carried out as described in *Darfeuille et al. (2007)* using PCR amplified fragments.

## Bioinformatic and statistical NGS data analyses

### Read pre-processing and alignment

Reads were first trimmed of low-quality 3' ends using cutadapt 1.1 (https://cutadapt.readthedocs.org/) and a base quality threshold of 28 (option '-q 28'). Then, reads having an average base quality lower than 28 were discarded using prinseq-lite 0.20.4 ('-min_qual_mean 28'; *Schmieder and Edwards, 2011*). Read pairs for which both mates passed the quality filtering steps were recovered by means of cmpfastq (http://compbio.brc.iop.kcl.ac.uk/software/cmpfastq.php), and mates were assembled into a single sequence using PANDAseq 2.9 (*Masella et al., 2012*) run with options '-N -o 30 -O 0 t 0.6 -A simple_bayesian -C empty'. About 5 million read pairs (combining the three biological replicates) could be assembled for the WT (*aapA3*/IsoA3) and pIsoA3* (*aapA3*/pIsoA3*) samples. These assembled reads were aligned onto the 426-nt reference sequence from *H. pylori* 26695 by the BWA-SW algorithm of BWA 0.7.12 (*Li and Durbin, 2009*) run with options '-a 1 -b 3 -q 5 r 2 -z 1' to produce alignments in BAM format. Mapped sequences of length 426 showing a single substitution compared to the reference were then extracted using utilities from the samtools 1.2 (*Li et al., 2009*) and bamtools 2.3.0 (*Barnett et al., 2011*) packages based on the various flags and tags in the BAM files (in particular the CIGAR string and NM tag). This gave a dataset of 1,653,406 WT and 2,559,164 pIsoA3* single-substitution sequences. Mapped sequences of length 425 and 427 harboring a single deletion or insertion, respectively, were also extracted (40,998 WT and 100,754 pIsoA3* single-deletion sequences; 4,799 WT and 7048 pIsoA3* single-insertion sequences).

## Statistical analysis

Statistical analyses of the differential distribution of substitutions in the WT and pIsoA3* single-substitution sequences were carried out. To determine whether substitutions were enriched at particular positions in the pIsoA3* compared to WT sequences, a 'positional' analysis was conducted by summing together the counts of all sequences that showed a substitution at a given position, regardless of the identity of the substituted nucleotide. A 'nucleotide-specific' analysis comparing the amount of each individual sequence was also done to determine whether particular nucleotides were enriched at specific positions. As positions +87 and +90 were mutated to inactivate the IsoA3 promoter (see *Figure 1B*), for the 'nucleotide-specific' analysis all sequences showing a difference to the reference at one or both of these two positions were excluded from the pIsoA3* and WT datasets (11,319 pIsoA3* and 38,575 WT sequences, respectively), and the pIsoA3* reference was converted back to the WT reference in order to make data comparable between WT and pIsoA3* samples. Differential analyses were conducted following the protocol of *Haas et al. (2013)* using tools from the Trinity 2.2.0 (*Haas et al., 2013*) and DESeq2 1.10.1 packages (*Love et al., 2014*), taking into account variability among the three biological replicates. The sequence abundance estimation step was not performed; actual sequence counts were used. The four positions at each extremity of the 426-nt amplicon corresponding to pieces of the primers could not be included in the statistical analyses as there were no substitutions at these positions in any the WT and pIsoA3* replicates. Substitutions were considered as significantly over- or under-represented in the pIsoA3* vs. WT samples if the p-value adjusted for multiple testing (False Discovery Rate [FDR] calculated using the Benjamini-Hochberg [BH] method in DESeq2) was equal or lower than 5% (padj $\leq$0.05). Bar plots of normalized sequence counts and log2 ratios of fold change were drawn using R 3.2.0 (R Core Team, 2015. R: A language and environment for statistical computing. R Foundation for Statistical Computing, Vienna, Austria; http://www.R-project.org/). Similar 'positional' analyses were also carried out for single-insertion and single-deletion sequences to determine whether insertions or deletions were statistically enriched at particular positions in the pIsoA3* dataset.

For the 'positional' analysis, a heatmap and hierarchical tree clustering of samples according to sequence count patterns was also performed. This was based on TMM-normalized (trimmed mean of M values), median-centered, log2-transformed FPKM (fragment per kilobase per million reads mapped) values, computed according to the protocol and tools of *Haas et al. (2013)*. The Pearson correlation coefficient was used as distance metric and average linkage was chosen as clustering method (options '–sample_dist sample_cor –sample_cor pearson –sample_clust average' for the 'analyze_diff_expr.pl' utility script). A log2 cut-off of 0 and a p-value cut-off of 1 were set (options '-C 0 P 1') in order to include all sequence positions in the map. The clustering script 'PtR' was

manually edited to suppress the clustering by sequence (i.e. rows) and sort the positions by numerical order instead.

Depending on the type of statistical analysis, 'position-specific' or 'nucleotide-specific', there were statistically enriched substitutions in absence of antitoxin (padj ≤0.05) at 70 or 72 positions within the *aapA3*/IsoA3 locus, respectively (65 positions in common between the two analyses). Substitutions identified in only one of the analyses included (relative to *aapA3* +1 TSS): i) positions −26 and −7 within the promoter region; ii) position +28 in the 5' UTR; iii) positions +64 and +97 in the AapA3 ORF; and iv) positions +146 and +177 in the 3' UTR. Such positions had generally a close-to-cut-off padj value, but not in all cases. For instance, position +28, which has been studied herein, had a highly significant padj value of $7.2 \times 10^{-6}$.

## Structural alignment and covariation analysis

To obtain further support for the proposed secondary structures, a phylogenetic analysis of nucleotide covariation in sequence alignments was carried out. A set of 107 AapA3 mRNA sequences from 55 *Helicobacter* genomes available in our T1TAdb database (https://d-lab.arna.cnrs.fr/t1tadb) was retrieved. Significantly mutated sequences (annotated as 'ghost' or with a '*' in T1TAdb) were not considered. In addition, strains corresponding to different isolates of other strains were discarded to avoid redundancy (i.e. Rif1 and Rif2, corresponding to 26695, BM012B and BM012S, corresponding to BM012A, and BM013B corresponding to BM013A). Using MAFFT (*Katoh and Toh, 2008*), a structural alignment was generated for two different transcript lengths (45 and 80 nucleotides) representing respectively the nascent AapA3 transcript extending up to the metastable structure 1 or 2. MAFFT was run in the 'X-INS-I' mode ('mafft-xinsi') with pairwise structural alignments computed by MXSCARNA (*Tabei et al., 2008*) ('–scarnapair' option) and a maximum of 1000 iterations ('–maxiterate 1000' option). Alignments were then manually slightly corrected. We used the secondary structures of AapA3 subsequences determined in vivo by DMS footprinting (*Figure 8*), covariation was revealed by means of R-chie (*Lai et al., 2012*). Options '–rule1=7' (to obtain base-pair covariation scores) and '–group1=4' (to group covarying base-pairs into four classes of scores) were set for R-chie.

## Data availability

The deep-sequencing raw and analyzed datasets reported in this paper have been deposited in the National Center for Biotechnology Information Gene Expression Omnibus (NCBI GEO) data repository under the accession code GSE121423 and can be accessed at the URL: https://www.ncbi.nlm.nih.gov/geo/query/acc.cgi?acc=GSE121423.

## Acknowledgements

We thank all present and past members of the ARNA laboratory for helpful discussions and in particular, Anaïs Le Rhun for critical reading of the manuscript. This work was supported by INSERM U1212, CNRS UMR 5320, Université de Bordeaux, and Agence Nationale de la Recherche (http://www.agence-nationale-recherche.fr/) grants Bactox1 and asSUPYCO. This work was performed in collaboration with the GeT core facility, Toulouse, France (http://get.genotoul.fr), and was then supported by France Génomique National infrastructure, funded as part of 'Investissement d'avenir' program managed by Agence Nationale pour la Recherche (contract ANR-10-INBS-09). This project has also received funding from the European Union's Horizon 2020 research and innovation programme under the Marie Sklodowska-Curie grant agreement No 642738. The funders had no role in study design, data collection and analysis, decision to publish, or preparation of the manuscript.

## Additional information

### Funding

| Funder | Grant reference number | Author |
|---|---|---|
| Agence Nationale de la Recherche | ANR-12-BSV5-0025-Bactox1 | Sandrine Chabas<br>Isabelle Iost<br>Fabien Darfeuille |

| H2020 Marie Skłodowska-Curie Actions | 642738 | Sara Masachis<br>Fabien Darfeuille |
| Agence Nationale de la Recherche | ANR-12-BSV6-0007-asSUPYCO | Sandrine Chabas<br>Isabelle lost<br>Fabien Darfeuille |
| Institut National de la Santé et de la Recherche Médicale | U1212 | Sara Masachis<br>Nicolas J Tourasse<br>Marion Faucher<br>Sandrine Chabas<br>Isabelle lost<br>Fabien Darfeuille |
| Centre National de la Recherche Scientifique | UMR 5320 | Isabelle lost<br>Fabien Darfeuille |

The funders had no role in study design, data collection and interpretation, or the decision to submit the work for publication.

## Author contributions

Sara Masachis, Conceptualization, Data curation, Validation, Investigation, Visualization, Methodology, Writing—original draft, Writing—review and editing; Nicolas J Tourasse, Resources, Data curation, Software, Formal analysis, Validation, Visualization, Methodology, Writing—original draft, Writing—review and editing; Claire Lays, Marion Faucher, Data curation, Validation, Investigation, Visualization, Methodology; Sandrine Chabas, Resources, Validation, Investigation, Methodology; Isabelle lost, Conceptualization, Supervision, Funding acquisition, Investigation, Methodology, Writing—original draft, Writing—review and editing; Fabien Darfeuille, Conceptualization, Resources, Formal analysis, Supervision, Funding acquisition, Investigation, Visualization, Methodology, Writing—original draft, Project administration, Writing—review and editing

## Author ORCIDs

Sara Masachis (iD) https://orcid.org/0000-0001-9968-0103
Isabelle lost (iD) https://orcid.org/0000-0001-9061-4177
Fabien Darfeuille (iD) https://orcid.org/0000-0003-1167-6113

## Decision letter and Author response

Decision letter https://doi.org/10.7554/eLife.47549.038
Author response https://doi.org/10.7554/eLife.47549.039

## Additional files

### Supplementary files

• Transparent reporting form
DOI: https://doi.org/10.7554/eLife.47549.034

### Data availability

Sequencing data have been deposited in GEO under accession code GSE121423.

The following dataset was generated:

| Author(s) | Year | Dataset title | Dataset URL | Database and Identifier |
| --- | --- | --- | --- | --- |
| Masachis S, Tourasse NJ, Lays C, Faucher M, Chabas S, lost I, Darfeuille F | 2018 | High-throughput suppressor selection of a type I toxin-antitoxin system in Helicobacter pylori | https://www.ncbi.nlm.nih.gov/geo/query/acc.cgi?acc=GSE121423 | NCBI Gene Expression Omnibus, GSE121423 |

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
