## [Decision Letter]

Thank you for submitting your article "A genetic selection reveals functional metastable structures embedded in a toxin-encoding mRNA" for consideration by *eLife*. Your article has been reviewed by three peer reviewers, and the evaluation has been overseen by Gisela Storz as the Senior and Reviewing Editor. The following individuals involved in review of your submission have agreed to reveal their identity: Kenn Gerdes (Reviewer #1) and Irmtraud Meyer (Reviewer #3).

The reviewers have discussed the reviews with one another and the Reviewing Editor has drafted this decision to help you prepare a revised submission.

Summary:

Masachis, Darfeuille et al. analyse a type I toxin – antitoxin (TA) module of the major human gastric pathogen Helicobacter pylori (Hp). Expression of toxins encoded by Type I modules is controlled by small, labile, cis-encoded antisense RNAs and often also by complicated mRNA metabolism that involves conserved mRNA folding pathways and/or mRNA processing. Using a combination of elegant and robust in vitro and in vivo methods, the authors first show that the aapA3/IsoA3 TA system of Hp is regulated in a way very similar to that of the homologous aapA1/IsoA1 system from the same organism (Figures 1 and 2). This initial part of the manuscript sets the stage for the next step, where the authors employ a powerful genetic screen combined with deep sequencing to identify single nucleotide changes that abolish production of the AapA3 toxin (Figure 3). This principle, which was invented by the authors, is technically robust, intellectually attractive and very powerful, and may yield novel insights that at present cannot be reached by other approaches. In particular, the authors discover that single point mutations outside the toxin gene reading frame suppress toxin gene translation. Focusing on the translation initiation region, they discover two mRNA hairpin structures that, when stabilized by single base changes, reduce translation by preventing ribosome binding (Figures 4 – 6). They propose that these structures are metastable and form during transcription to keep the toxin translation-rate low, as explained in the model figure (Figure 7).

Essential revisions:

All of the reviewers thought the quality of the experimental work in the manuscript is outstanding and the conclusions are justified. However, all thought it would be nice to have additional evidence of the proposed metastable structures in the nascent toxin mRNA. While the reviewers understood this might be technically difficult, they agreed that it is worth a try and had the following suggestions.

1) Phylogeny (i.e. nucleotide co-variation in sequence alignments) was previously used to deduce the existence of stem-loop structures not only in ribosomal RNAs but also in mRNAs (e.g., hok mRNAs). Did the Authors consider using this approach to support the existence of the proposed metastable structures in the nascent toxin transcript? This possibility depends on the actual homologous sequences available and is not possible in all cases. If phylogeny indeed supports the existence of the metastable structures, the Authors could look for coupled nucleotide covariations that would support a conserved mRNA folding pathway (that is, one mRNA sequence elements pairs with two or more other elements during the fife-time of the mRNA). The authors state in the Discussion that "these local hairpins were previously predicted to form during the co-transcriptional folding pathway of several AapA mRNAs (Arnion et al., 2017)." However, they authors did not explain how these hairpins were predicted. It is worth explaining this central point.

2) Although transient structures are by definition hard to detect, the authors could try in vivo structure probing (DMS) of truncated mRNAs 1-64 and 1-90 to demonstrate the existence of the first and the second metastable structures, respectively.

3) It is preferable to carry out 2D structure predictions on the naturally occurring transcript, not a sub-sequence. 2D structure prediction generated by algorithms such as RNAfold (or Mfold) that are guided by δ-G stability optimisation are sensitive to the sequence context, so the correct sequence needs to be used to be able to draw conclusions. Additionally, the findings presented in Figure 3D could be analyzed a bit further to produce significant, independent evidence for some structure features. Specifically:

Figure 2 caption:

- – "2D structure predictions were generated with the RNAfold Web Server (Gruber, Lorenz, Bernhart, Neuböck, and Hofacker, 2008) and VARNA (Darty, Denise, and Ponty, 2009) was used to draw the diagrams."

- Please state clearly whether any of the results of the experimental 2D structure probing were used as input to RNAfold (i.e. as additional constraints to the prediction algorithm).

Figure 3D:

- Please add coloring to the peaks depending on which codon position they overlap (1, 2 or 3) and carefully discuss the corresponding results, also in the context of the 2D structure elements.

- Given that you have a decent number of pair-mutations, analyze them to see whether any correspond to RNA structure base-pairs (and whether any of the pair mutations rescue the base-pair and thus affect the system differently). This would serve as additional, independent evidence of 2D structure probing and predictions.

---

## [Author Response]

Essential revisions:All of the reviewers thought the quality of the experimental work in the manuscript is outstanding and the conclusions are justified. However, all thought it would be nice to have additional evidence of the proposed metastable structures in the nascent toxin mRNA. While the reviewers understood this might be technically difficult, they agreed that it is worth a try and had the following suggestions.1) Phylogeny (i.e. nucleotide co-variation in sequence alignments) was previously used to deduce the existence of stem-loop structures not only in ribosomal RNAs but also in mRNAs (e.g., hok mRNAs). Did the Authors consider using this approach to support the existence of the proposed metastable structures in the nascent toxin transcript? This possibility depends on the actual homologous sequences available and is not possible in all cases. If phylogeny indeed supports the existence of the metastable structures, the Authors could look for coupled nucleotide covariations that would support a conserved mRNA folding pathway (that is, one mRNA sequence elements pairs with two or more other elements during the fife-time of the mRNA). The Authors state in the Discussion that "these local hairpins were previously predicted to form during the co-transcriptional folding pathway of several AapA mRNAs (Arnion et al., 2017)." However, they authors did not explain how these hairpins were predicted. It is worth explaining this central point.

Following the reviewers’ suggestion, we used a phylogenetic approach to analyze nucleotide co-variation in sequence alignments. For this analysis, we used 107 AapA3 mRNA sequences from 55 *Helicobacter* genomes available in our T1TAdb database (https://d-lab.arna.cnrs.fr/t1tadb). A structural alignment was carried out on two different transcript lengths (around 45 and 80 nucleotides) using MAFFT (Katoh and Toh, 2008), and then manually corrected. Covariation was revealed by means of R-chie (Lai et al., 2012). This analysis supports the secondary structure determined in vivo (see point 2 below) for both nascent mRNAs. The existence of both MeSt1 and MeSt2 metastable hairpins is very well supported by the nucleotide co-variation observed. These results are now presented in Figure 8—figure supplement 1 and discussed in the corresponding section. A complete and detailed description of this analysis can be found in the Materials and methods section.

Regarding the prediction of the metastable hairpins during co-transcriptional folding in our previous work, this was done using the KineFold software with default parameters (Xayaphoummine et al., 2005). This precision has now been included in the text.

*2) Although transient structures are by definition hard to detect, the authors could try* in vivo *structure probing (DMS) of truncated mRNAs 1-64 and 1-90 to demonstrate the existence of the first and the second metastable structures, respectively.*

To confirm the existence of the metastable structures, we used in vivo DMS footprint analysis, as suggested by the reviewers. We constructed two *H. pylori* strains that constitutively express the 1-60 or 1-90 aapA3 transcripts at the endogenous *aapA3* chromosomal locus. These truncated transcripts are expected to mimic the two nascent RNAs harboring the first and second metastable structure, respectively. These short transcripts were fused upstream of the well-characterized non-coding RNA RepG to ensure both their stability and a proper transcription termination. Northern blot analysis confirmed that these chimeric transcripts were stable and displayed a defined size (data not shown). We also constructed variant strains containing the A40T and T78C suppressor mutations that respectively stabilize the MeSt1 and MeSt2 structures. The four strains were treated with DMS and methylated nucleotides were detected by reverse transcription. As shown in Figure 8, the pattern of modification obtained fits very well with the predicted secondary structures, confirming that both MeSt1 and MeSt2 hairpins can form in vivo. These results definitely reinforce the conclusions of our work and they have now been included in the Results section (Figure 8).

3) It is preferable to carry out 2D structure predictions on the naturally occurring transcript, not a sub-sequence. 2D structure prediction generated by algorithms such as RNAfold (or Mfold) that are guided by δ-G stability optimisation are sensitive to the sequence context, so the correct sequence needs to be used to be able to draw conclusions. Additionally, the findings presented in Figure 3D could be analyzed a bit further to produce significant, independent evidence for some structure features. Specifically:Figure 2 caption:- "2D structure predictions were generated with the RNAfold Web Server (Gruber, Lorenz, Bernhart, Neuböck, and Hofacker, 2008) and VARNA (Darty, Denise, and Ponty, 2009) was used to draw the diagrams."- Please state clearly whether any of the results of the experimental 2D structure probing were used as input to RNAfold (i.e. as additional constraints to the prediction algorithm).

We have now modified the text in the legends of Figures 2, 5 and 6 to clearly specify whether additional constraints were used to predict the secondary structures. Moreover, we are also specifying that the structures shown in Figures 5 and 6 are snapshots of longer structures.

Figure 3D:- Please add coloring to the peaks depending on which codon position they overlap (1, 2 or 3) and carefully discuss the corresponding results, also in the context of the 2D structure elements.

We have now modified Figure 3D by adding a specific color code for each codon position (1, 2 and 3) and discussed these results in the Results section.

- Given that you have a decent number of pair-mutations, analyze them to see whether any correspond to RNA structure base-pairs (and whether any of the pair mutations rescue the base-pair and thus affect the system differently). This would serve as additional, independent evidence of 2D structure probing and predictions.

We have added a figure showing the location of all single mutations in non-coding regions on the secondary structure of the AapA3 mRNA (Figure 3—figure supplement 3). As pointed out by the reviewer, most of these mutations disrupt base-pairing within two stem-loop structures. A more detailed analysis of these mutations will be presented in a future manuscript.